# From Real to Synthetic: A Fine-grained Dataset and High-fidelity Biomechanical Model for Animal Behavior Understanding

## Abstract

Rat behavior research contributes to the exploration of human disease mechanisms. However, existing datasets are scarce and cover limited behavior types, hindering the analysis and modeling of complex behavior patterns. We constructed ActionRat, a new multi-view rat behavior dataset that, for the first time, captures diverse actions during free exploration and brain-computer interface (BCI) control. It combines real and synthetic sequences with fine-grained keypoint annotations and atomic action sequences, supporting broader behavior analysis tasks. To efficiently generate synthetic data for dataset expansion, we developed OpenRatEngine, a high-fidelity 3D virtual biomechanical model. This model integrates anatomical priors from computed tomography (CT) scans, kinematic constraints, and lifelike appearance, reducing the domain gap between synthetic and real data. Equipped with pose control, OpenRatEngine generates synthetic sequences with accurate 3D keypoint annotations. We evaluated behavioral uncertainty quantification and animal pose estimation tasks on the ActionRat dataset, and demonstrated the outstanding synthetic data generation capability and realism of OpenRatEngine. Extensive experiments across deep learning models confirmed the effectiveness and value of both real and synthetic data.

## 1 Introduction

Rats share significant anatomical, physiological, genetic, and behavioral similarities with humans, and are therefore commonly used as experimental subjects for studying human diseases, such as Parkinson's disease (Li et al., 2023) and autism (Klibaite et al., 2025). Tracking the positions of gait, limbs, and torso of animals is essential for reliably identifying normal and pathological patterns. This has advanced animal behavior quantification and pose estimation algorithms (Ye et al., 2024; Nath et al., 2019), improving analysis efficiency. However, algorithm performance remains dependent on diverse, large-scale, and well-annotated datasets (Sheppard et al., 2022), whereas rat behavioral data are scarce.

Previous public animal behavior datasets have primarily focused on 2D pose estimation from single-view videos (Khan et al., 2020; Yu et al., 2021). Occlusions and shape variations often result in missing or blurred action details, limiting the accurate motion characterization and reducing fine-grained behavior recognition. While some studies have introduced 3D pose information via marker-based motion capture systems that improve precision, these methods are costly and marker interference with animals' natural movements (Bala et al., 2020; Marshall et al., 2021). Recent efforts have explored multi-view recordings, such as Rodent3D (Patel et al., 2023) and Rat7M (Dunn et al., 2021) datasets, with calibrated cameras and triangulation to reconstruct 3D poses. Nevertheless, these datasets mainly capture common behaviors during spontaneous exploration and lack atypical behaviors induced by interventions or pathology, leading to limited behavioral diversity.

In addition, to alleviate data scarcity and annotation difficulties, researchers increasingly employ synthetic data as a complement. PASyn (Jiang et al., 2022) learns quadruped pose priors and applies Blender to create the SynAP dataset. Recently, the AP-CAP pipeline (Wang et al., 2025) integrates a multi-modal image generation model with caption enhancement to fuse real and synthetic data, producing the MPCH dataset for animal 2D pose estimation. For rodents, synthetic methods have

Table 1: Comparison of animal datasets. "APE": animal pose estimation, "AAR": animal action recognition, "AR": animal reconstruction, "SDG": synthetic data generation, "ASE": animal shape estimation, "VG": video grounding, "UQ": uncertainty quantification, "B. Box": bounding box, "Seg. M": segmentation maps, and "SMAL": skinned multi-animal linear model. Our ActionRat dataset integrates comprehensive sources, comprising multi-camera real laboratory recordings, 2D and 3D keypoints, action category (Cat.) annotations, rat CT scans, synthetic (Syn.) sequences, and logs of electrical stimulation instructions (ESI) based on brain-computer interface (BCI).

| Dataset | Species | Frames | Annot. Method | Annotations | Keypoints | Source | Task |
|---|---|---|---|---|---|---|---|
| OpenMonkeyStudio (Bala et al., 2020) | Macaque | 195K | Marker-based | 2D, 3D | 13 | Real | APE, AAR |
| Rat7M (Dunn et al., 2021) | Rat | 7M | Marker-based | 3D | 20 | Real | APE |
| AP-10K (Yu et al., 2021) | Wild animals | 59K | Manual | 2D | 17 | Real | APE |
| AcinoSet (Joska et al., 2021) | Cheetah | 119K | Manual | 2D, 3D | 20 | Real | APE |
| MARS (Segalin et al., 2021) | mouse | 1.5M | Manual | 2D | 7 | Real | APE, AAR |
| Animal Kingdom (Ng et al., 2022) | Animals | 4.3K-33K | Manual | 2D, Cats. | 23 | Real | VG, AAR, APE |
| TriMouse-161 (Lauer et al., 2022) | Mouse | 11K | Manual | 2D | 12 | Real | APE |
| Animal3D (Xu et al., 2023) | Mammal | 3.4K | Auto., Manual | 3D, Shape | 26 | Real | APE, ASE |
| Rodent3D (Patel et al., 2023) | Rodent | 4.5M | Tri., Manual | 2D, 3D | 8 | Real | APE |
| TopViewMouse-5k (Ye et al., 2024) | Mouse | 5K | Manual | 2D | 26 | Real | APE |
| SyDog-Video (Shooter et al., 2024) | Dog | 87.5K | Auto. | 3D, B. Box, Seg. M | 33 | Syn. | APE |
| GenZoo (Niewiadomski et al., 2024) | Mammal | 1M | Auto. | 3D, Shape | 35 | Syn. | APE, ASE |
| CtrlAni3D (Lyu et al., 2025) | Animals | 9.7K | Auto. | 2D, 3D, SMAL | 35 | Syn. | APE, AR |
| MPCH (Wang et al., 2025) | Animals | 164K | Auto., Manual | 2D | 17, 23 | Real, Syn. | APE |
| **ActionRat (Ours)** | Rat | 609K | Auto., Manual | **2D, 3D, ESI, Cats.** | **12, 60** | **CT, Real, Syn.** | **APE, UQ, SDG** |

been applied to mice (Bolaños et al., 2021; Sosa et al., 2023; Meijer et al., 2021), while high-quality synthetic data for rat behaviors remains largely unexplored in the vision community.

The challenge of rat behavior construction remains an unresolved issue. (i) For real-world data, existing multi-view datasets lack recordings of critical abnormal behaviors, such as convulsions and spasms, limiting model performance in behavior quantification and recognition tasks. (ii) For synthetic data, simulated rats differ significantly from real ones in appearance, morphology, and dynamics, resulting in a domain gap that makes it difficult to accurately replicate natural motion sequences and hinders the transferability of models trained on synthetic data (Li & Lee, 2021). Given these issues, how should AI developers and researchers build a behaviorally diverse dataset to support model generalization and robustness? Furthermore, how should they improve the reliability, realism, and efficiency of synthetic data generation?

These observations inspired us to propose ActionRat, a multi-source synchronized dataset focusing on rat behavior, and Table 1 summarizes its core characteristics: (i) Superior in diversity. Unlike existing datasets such as Rat7M (Dunn et al., 2021), which are limited to spontaneous behaviors during free exploration, ActionRat additionally introduces numerous BCI-controlled behaviors that differ markedly from natural patterns. These intervention actions, driven by intrinsic intentions and external stimulation, often exhibit limb discoordination, abrupt turns, brief rhythms, or rigid postures. By integrating both free and intervention behaviors with a combination of real and synthetic data, ActionRat greatly enhances behavioral diversity and achieves comprehensive coverage of motion patterns. (ii) Superior in quality. ActionRat is a well-curated dataset for animal behavior understanding tasks such as behavioral uncertainty quantification and 3D pose estimation. It comprises 609K multi-view frames, with synthetic sequences annotated with 60 3D keypoints, real sequences with 12 keypoints, and corresponding BCI electrical stimulation instructions (ESI) logs. Unlike previous datasets, we define seven semantically grounded atomic actions for rats: locomotion, turning, sniffing, pausing, rearing, grooming, and micro-movements. Any motion sequence can be decomposed into a composition of these units, for example a trajectory from A to B may be represented as "locomotion→turning→locomotion→pausing".

Furthermore, we created OpenRatEngine, a virtual rat model combining anatomical and kinematic priors to generate high-fidelity synthetic data. The skeletal structure was reconstructed from high-resolution computed tomography (CT) scans, and a detailed skin mesh preserves external morphology and captures fine body contours. This model enables the generation of naturalistic behavior sequences and effectively addresses occlusion issues in 3D pose estimation. Compared to existing rodent biomechanical models (Aldarondo et al., 2024), OpenRatEngine demonstrates superior performance in skeletal anatomical accuracy, biomimetic appearance, and behavioral expressiveness. It is capable of capturing and synthesizing fine-grained variations, including sniffing and micro-movements, thereby exhibiting higher fidelity in dynamic detail reproduction.

The main contributions are as follows:

- ActionRat serves as an extensive multi-purpose animal dataset with precise keypoint and fine-grained action annotations. To the best of our knowledge, it is the first available dataset including both real and synthetic data, covering both free and intervention behaviors.

- We developed OpenRatEngine, a high-fidelity 3D virtual rat biomechanical model that integrates anatomical priors, inverse kinematics constraints, and a lifelike appearance to generate structurally complete synthetic data with 3D keypoint annotations while minimizing the domain gap with real data (refer to the supplementary video).

- We conducted benchmark evaluations across behavioral uncertainty quantification and 3D pose estimation tasks. Extensive studies demonstrate the superior performance of the ActionRat dataset and the OpenRatEngine model.

## 2 RELATED WORK

### 2.1 ANIMAL BEHAVIOR DATASETS

Recent advances in pose estimation and multi-camera technologies have expanded animal behavior datasets, enhancing spatial accuracy and behavioral semantics.

**Single-view datasets** are primarily constructed from images of individual animals, from which kinematic data such as head position and orientation are extracted using 2D and 3D pose estimation algorithms. Representative approaches include LEAP (Pereira et al., 2019), DeepLabCut (Mathis et al., 2018), and DeepPoseKit (Graving et al., 2019). However, occlusions and viewpoint limitations often degrade animal keypoint accuracy and potential drift, hindering reliable behavioral dynamics reconstruction. **Multi-view datasets** employ synchronized cameras to reconstruct 3D skeletal trajectories of animals with improved spatial precision (Joska et al., 2021), using either convolutional networks or triangulation (Xu et al., 2023). In addition, some multimodal datasets (Dunn et al., 2021) incorporate wearable sensors, equipping animals with retroreflective markers on the head, trunk, and limbs for 3D behavior analysis using motion capture systems. Nevertheless, these efforts mainly emphasize spatial accuracy, often neglecting behavioral semantics. **Semantic-enhanced datasets** place greater emphasis on providing high-semantic-density labels and event-level annotations to support behavior quantification and recognition (Lyu et al., 2025). The Animal Kingdom dataset (Ng et al., 2022) offers large-scale, diverse animal behavior data, but lacks dedicated rodent-specific organization. MARS (Segalin et al., 2021) and SLEAP (Pereira et al., 2022) focus on mouse social behaviors, offering multi-animal tracking and annotations. The BamaPig3D dataset (An et al., 2023) employs the MAMMAL system to achieve surface mesh reconstruction under occlusions, enabling the recovery of complete body postures for social behavior recognition in pigs. In addition, a multimodal dataset (Patel et al., 2023) combining thermal infrared and RGB-D cameras synchronously record single-rat and paired-rat behaviors during naturalistic exploration and foraging. However, rat-specific datasets remain limited in scale and action diversity, hindering in-depth analysis of behavioral differences.

Most existing public datasets primarily focus on pose estimation or behavior recognition, and they contain only common actions. To address this, we collected extensive multi-view synchronized videos of rats exhibiting both exploratory and intervention-induced behaviors, creating a dataset with richer semantics and broader behavioral coverage to support a wider range of research tasks.

### 2.2 ANIMAL SYNTHETIC DATA GENERATION

Computational modeling has advanced animal behavior research, with virtual reconstruction and synthetic data enabling realistic motion simulation, large-scale annotation, and more robust analysis.

**Virtual animal modeling** aims to reconstruct both the anatomical structures and motor behaviors of animals. Early studies mainly relied on geometry-based modeling (Ilett et al., 2023), in which motion generation was driven by learned trajectories. However, such approaches lack dynamical constraints and struggle to ensure physical consistency when applied to new behaviors. In recent years, physics-based modeling has emerged as a promising approach, simulating dynamic interactions between body components and the environment. For instance, Aldarondo (Aldarondo et al., 2024)

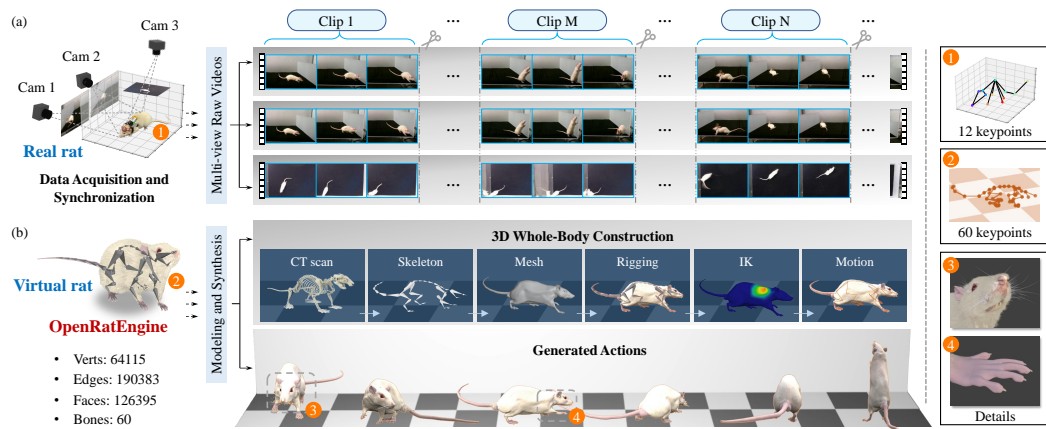

Figure 1: Real data acquisition and synthetic data generation. (a) Real data were synchronously captured by multi-view cameras, covering free rat motion and BCI-controlled movements. (b) Synthetic data were generated by the virtual rat model OpenRatEngine, featuring a CT-based skeletal structure, biomimetic mesh appearance, and inverse kinematics (IK) constraints.

developed a virtual rodent biomechanical model driven by an artificial neural network to investigate how the brain controls movement. Victor et al. (Lobato-Rios et al., 2022) proposed a neuromechanical model of adult Drosophila melanogaster to explore how neural dynamics, musculoskeletal properties, and the environment jointly shape behavior. Compared with geometric approaches, physics-based methods can generate motion that is both biologically plausible and interpretable. Moreover, **synthetic data** has emerged as a supplementary strategy to address the challenges of data scarcity and high annotation costs in animal behavior analysis, with its feasibility demonstrated in various studies (Shooter et al., 2024). Several simulation models have been developed, such as replicAnt (Plum et al., 2023) and virtual mouse (Bolaños et al., 2021). By controlling virtual animals, these models can produce large-scale, multi-view synthetic images and corresponding annotations for model training. To further bridge the domain gap between synthetic and real data, recent research has introduced techniques such as domain randomization (Mu et al., 2020) and adversarial domain adaptation (Li & Lee, 2021) to enhance the transferability of synthetic data to real-world tasks.

Inspired by these advances, we developed a high-fidelity 3D rat model, capable of simulating realistic rat behavior patterns and generating training data for both supervised pose estimation and behavioral quantification methods.

## 3 THE ACTIONRAT DATASET AND OPENRATENGINE

The proposed ActionRat dataset comprises synchronized multi-view recordings of six rats during free exploration and behaviors controlled by BCI-driven electrical stimulation. It includes raw videos, synthetic sequences, 2D/3D keypoint annotations, camera calibration parameters, and the corresponding codebase.

### 3.1 REAL DATA ACQUISITION AND SYNCHRONIZATION

As previously discussed, existing rat datasets are scarce and have coarse-grained annotations, hindering in-depth evaluation of behavioral patterns. Therefore, we designed a dedicated experimental setup and systematically collected a new rat dataset with fine-grained annotations and diverse behavioral coverage, as illustrated in Fig. 1(a).

In the first stage, multi-view videos were synchronously recorded using a binocular (camera 1 and camera 2) and a monocular camera (camera 3), mounted on the side and top of the arena, respectively. Prior to recording, the binocular camera was calibrated with checkerboard patterns using OpenCV, ensuring spatial accuracy and viewpoint consistency. Videos were captured at 1920×1080 pixels and 30 frames per second (FPS). We selected six eligible adult male Sprague-Dawley rats from 10 candidates. The first round recorded free exploration of six rats. In the second round, rats

were fitted with wearable devices that used BCI technology to deliver electrical stimulation guiding forward and turn movements. Due to individual variability, stimulation may induce abnormal behaviors such as spasms, collisions, spin, dash, fall, and limb incoordination—all recorded. This stage yielded a long-term video dataset comprising 609K frames of both normal and abnormal behaviors (referred to as Version 1) along with corresponding ESI logs. During the second stage, we reviewed the V1 recordings and segmented them into 64-frame clips according to seven atomic units. The resulting dataset, termed Version 2 (V2), consists of 8679 clips with extensive posture and motion variations, including 1665 locomotion, 1179 turning, 987 sniffing, 1863 pausing, 558 rearing, 501 grooming, and 1926 micro-movement. A detailed statistical comparison is provided in Appendix B.

**Ethical statement**. All experiments are conducted in accordance with the guidelines for the care and use of laboratory animals and approved by the Laboratory Animal Welfare and Ethics Committee.

### 3.2 OpenRatEngine Modeling and Motion Synthesis

To scale up data generation and produce high-quality synthetic training data, we developed OpenRatEngine, a high-fidelity virtual rat biomechanical model integrating anatomical priors, IK priors, and realistic appearance. Its motion control closely aligns with real rat behavior, effectively reducing the domain gap between synthetic and real data. The overall pipeline is illustrated in Fig. 1(b).

**Model definition**. We decouple the pose of OpenRatEngine into a static component $S$ and a dynamic structure $\mathbf{Q}$, used to represent posture and drive motion, respectively. The static structure $S$ is defined as:

$$S = \left\{ \mathbf{s}_i \in \mathbb{R}^3 \mid i = 1, ..., N \right\}, \tag{1}$$

where $N$ denotes the total number of joints in the skeleton. The spatial offset $\mathbf{s}_i$ (i.e., distance and direction) of joint $i$ relative to its parent node coordinates $\mathbf{p}_{\text{parent}(i)}$, is given by $\mathbf{s}_i = \mathbf{p}_i - \mathbf{p}_{\text{parent}(i)} \in \mathbb{R}^3$. Here, the parent node refers to the anatomically upstream joint that directly connects to joint $i$ in the skeletal hierarchy (refer to Appendix C.2 for more details). The dynamic structure $\mathbf{Q}$ characterizes the motion process by representing the temporal changes of each joint using rotation matrices $\mathbf{R}_i(t)$ and translations $\mathbf{d}_i(t)$. It is defined as:

$$\mathbf{Q}(t) = \left\{ (\mathbf{R}_i(t), \mathbf{d}_i(t)) \mid i = 1, \ldots, N \right\}, \tag{2}$$

where $\mathbf{R}_i(t) \in \text{SO}(3)$ describes the rotation of joint $i$ relative to its parent in the local coordinate system at frame $t$. This rotation is parameterized as a quaternion and subsequently converted into a rotation matrix. The global position of each joint is then recursively computed through forward kinematics along the parent–child hierarchy. The $\mathbf{d}_i(t) \in \mathbb{R}^3$ represents the positional displacement of the root joint in the world coordinate system. By combining the static structure $S$ with the dynamic structure $\mathbf{Q}(t)$, the 3D pose $\mathbf{P}_i(t)$ of each joint at any given time $t$ can be reconstructed:

$$\mathbf{P}_i(t) = \mathbf{R}_{\text{parent}(i)}(t) \cdot \mathbf{s}_i + \mathbf{d}_i(t) + \mathbf{P}_{\text{parent}(i)}(t). \tag{3}$$

The entire motion process can be represented as a set of joint poses spanning all time steps and skeletal nodes:

$$\mathcal{P} = \left\{ \mathbf{P}_i(t) \mid i = 1, \ldots, N; \ t = 1, \ldots, T \right\}, \tag{4}$$

where $T$ denotes the total number of frames in sequence, and $\mathcal{P}$ captures the complete trajectory in 3D space.

**Skeleton construction**. To bridge the gap between real and synthetic domains in limb motion reconstruction and to ensure biomechanical realism and accuracy in OpenRatEngine, we extracted the full skeletal structure of real rats from high-resolution CT scans. For computational efficiency, the skeleton was simplified into five regions: head, neck, spine, limbs, and tail. In each region, we retained joints critical for behaviors (e.g., scapula, lumbar vertebra, metacarpal bone) and merged or discarded bones with minimal contribution to movements (e.g., coccygeal vertebrae, ribs). This strategy reduces redundancy while maintaining anatomical consistency with the real skeleton. Skeletal segments were created in Blender to construct the OpenRatEngine skeleton, as shown in Fig. 1. The final structure preserves the principal locomotor components, comprising 60 bones and 82 vertices, and each bone is equipped with a local coordinate system anchored at its root joint. Detailed bone connectivity is provided in the Appendix C.2.

**Biofidelic appearance construction**. To mitigate the morphological gap between real and synthetic domains, we designed a lifelike appearance for OpenRatEngine, with details shown in Fig. 1. A

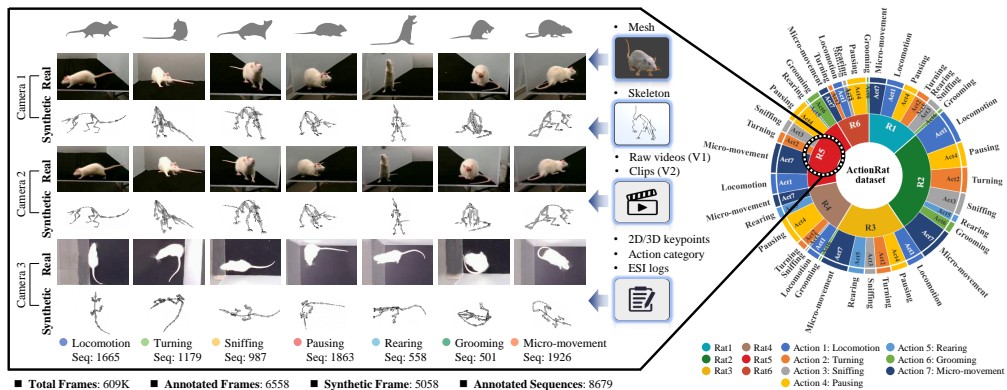

Figure 2: ActionRat is a multi-purpose dataset providing multi-view raw videos of real rat motion (V1) with the corresponding ESI logs, cropped atomic action clips (V2), and 3D synthetic sequences generated by OpenRatEngine. Annotations include 2D keypoints on real images, 3D keypoints of synthetic sequences, and action categories for each clip.

high-resolution mesh with 64115 vertices, 190383 edges, and 126395 faces formed the deformation basis. Skin, textures, and materials were added, and a particle system was used to simulate fine-grained fur. Visual features were modulated based on real Sprague-Dawley rat images to ensure appearance consistency, enhancing the model's realism and cross-domain transferability. Additionally, we constructed a controllable virtual open-field environment with adjustable noise, lighting, and camera angles.

**Manipulation control based on contour consistency**. To ensure flexible surface deformation during motion control, OpenRatEngine was rigged by mapping skeletal nodes to the mesh. Each node $i$ was assigned a vertex weighting function $\omega_i(u) : \mathcal{U} \to [0, 1]$, where $u \in \mathcal{U}$, $\mathcal{U}$ denotes the set of mesh vertices, and $\omega_i(u)$ represents the influence of node $i$ on vertex $u$. Higher weights produce stronger deformations. We manually assigned the initial weights (as in (Bolaños et al., 2021)) and iteratively refined through pose adjustments and weight optimization across diverse motion tests, thereby preventing surface collapse or protrusion under large deformations. We further incorporated IK to infer upstream joint motions from distal joints, enhancing motion coherence.

We propose a multi-view contour consistency pose control method to synthesize motion sequences. Given the real rat contours $C_k(t) \in \mathbb{R}^{H \times W}$ at frame $t$ from three camera views $k = 1, 2, 3$, the projected contour of OpenRatEngine under pose $\mathbf{P}(t)$ in the corresponding virtual view is denoted as $\mathcal{M}(\mathbf{P}(t), k)$. The optimization objective is defined as:

$$\min_{\Theta(t)} \sum_{k=1}^{3} \mathcal{D}\left(\mathcal{M}(\mathbf{P}(t), k), C_k(t)\right), \quad \text{s.t.} \quad \mathcal{F}_{\text{ground}}(\mathbf{P}(t)) \geq 0, \tag{5}$$

where $\mathcal{D}(\cdot)$ denotes the contour discrepancy metric, $\Theta(\cdot)$ denotes the set of rotation and translation parameters, and $\mathcal{F}_{\text{ground}}(\cdot)$ enforces ground-contact constraints to ensure foot-ground contact without penetration. Through a semi-automatic optimization process, we defined sparse keyframes in Blender to represent characteristic rat poses and applied interpolation to generate smooth transitions, enabling discrete skeletal keypoints to naturally extend into complete motion sequences and thereby improving the efficiency of synthetic data generation. The synthesized sequences include precise 3D keypoint annotations and cover diverse behaviors in the virtual open-field environment.

### 3.3 ANNOTATIONS AND STATISTICS OF THE ACTIONRAT DATASET

The ActionRat dataset includes real rat motion data and synthetic sequences generated by Open-RatEngine, with comprehensive annotations. We manually annotated 12 2D keypoints on 1500 real images for training pose estimation models such as DeepLabCut (Mathis et al., 2018; Nath et al., 2019). OpenRatEngine generated synthetic sequences by manipulating silhouettes from 5058 real images, with all bone heads defined as keypoints, resulting in 60 annotated 3D keypoints. It should be noted that real data provide relatively sparse annotations due to the limited visual resolution on

small target and occlusions, as referred to prior rodent datasets (8–26 keypoints), such as Rodent3D (Patel et al., 2023) and TriMouse-161 (Lauer et al., 2022). In contrast, Our OpenRatEngine automatically generates dense 3D skeletal annotations (60 keypoints), greatly enhancing annotation accuracy and completeness. Three biology experts provided fine-grained behavior labels for 8679 real and synthetic segments, covering atomic actions such as locomotion, turning, sniffing, pausing, rearing, grooming, and micro-movements. Three rounds of cross-validation ensured annotation accuracy and consistency. Labeled data were split into training, validation, and test sets at 8:1:1. ActionRat V2 offers clearer taxonomy and richer annotations than existing rat datasets, with statistical details provided in Appendix B.6. The ActionRat dataset and the virtual rat model OpenRatEngine will be released upon paper acceptance.

## 4 EXPERIMENTS

### 4.1 BEHAVIORAL UNCERTAINTY QUANTIFICATION

We conducted two comparative experiments on the ActionRat dataset against nine baseline models. ActionRat encompasses a rich variety of behavioral patterns, and results demonstrate its effectiveness in advancing research on animal behavior uncertainty quantification task.

**Experimental details**. We selected nine representative prediction models as baselines, including PoseMamba (Huang et al., 2025), Diffpose (Gong et al., 2023), siMLPe (Guo et al., 2023), Motionformer (Aksan et al., 2021), CCVAE (Cheng et al., 2020a), ABPLSTM (Roberts & Segev, 2020), MCENET (Cheng et al., 2020b), Social-GAN (Gupta et al., 2018), and STGCN (Yu et al., 2017). We comprehensively evaluated the performance of these models on three animal and one human datasets, including our ActionRat, Animal Kingdom (Ng et al., 2022), MARS (Segalin et al., 2021) and Penn Action (Zhang et al., 2013). All models take the keypoint sequences from the top-view perspective of the ActionRat V1 dataset as input. The predicted results were quantitatively evaluated using mean per joint position error (MPJPE), mean per joint velocity error (MPJVE), and Diversity metrics. Experiments were conducted on an NVIDIA RTX 3080 Ti (12GB) using PyTorch, employing Leaky ReLU (slope 0.1), dropout rate of 0.2, learning rate of 0.05, batch size of 64, 500 training epochs, and the AdamW optimizer.

**Result and analysis**. Table 2 reports MPJPE and MPJVE for nine baseline models across different datasets. Under identical models and training configurations, the prediction errors on ActionRat are systematically higher than on the other datasets in both pixels (px) and decimeters (dm), indicating that ActionRat constitutes a more challenging benchmark for behavioral uncertainty quantification. Specifically, in the left (px) of Table 2: (i) Across models, the best results typically occur on Penn Action or Animal Kingdom, whereas the ActionRat column almost always yields the largest errors. This suggests that existing models can already fit the motion patterns in Penn Action and Animal Kingdom reasonably well, but still suffer a clear performance drop when confronted with the more complex behavioral distribution in ActionRat. (ii) Within each dataset, performance consistently improves as the model becomes more advanced. Even for the strongest models, the errors on ActionRat remain noticeably higher than on the other datasets. For example, when moving from Penn Action to ActionRat, the MPJPE of ABPLSTM, MCENET, DiffPose, and PoseMamba increases by approximately 20%–35%.

The diversity results in Table 3 show that, under identical models and training configurations, the ActionRat column almost always attains the best or second-best Diversity scores across the nine baseline models, with diversity consistently exceeding that of MARS, Animal Kingdom, and Penn Action in both pixel space (pixels) and physical space (decimeters). This aligns with the behavioral control mechanisms in our setting: BCI stimulation parameters are strongly mapped to behavioral outcomes, while spontaneous movements are further modulated by external factors such as smell, lighting, and obstacles, yielding more complex and variable trajectories even within the same semantic action. Consequently, under the same evaluation protocol, ActionRat exhibits a broader range of motion patterns and higher motion diversity, which is consistent with the increased prediction difficulty observed in Table 2, and indicates that it is a more challenging benchmark.

To evaluate the effect of ESI labels as prior information in behavioral uncertainty quantification, we conduct comparative experiments between models with and without ESI input. The baseline model predicts future keypoints using only historical keypoint sequences, whereas the ESI-augmented vari-

Table 2: Behavioral uncertainty quantification task were conducted using nine baseline models. Left: Model performance on four datasets was assessed using MPJPE (↓) and MPJVE (↓) metrics **in pixels (px)**. Right: Model performance was evaluated on three datasets using the same metrics **in decimeters (dm)**. Best and second-best results are highlighted in bold and underlined, the gray-shaded ActionRat column denotes the most challenging dataset.

| Method | Penn Action | Animal Kingdom | MARS | ActionRat | Penn Action | Animal Kingdom | ActionRat |
|---|---|---|---|---|---|---|---|
| | MPJPE ↓ / MPJVE ↓ (px) | | | | MPJPE ↓ / MPJVE ↓ (dm) | | |
| ABPLSTM | 30.27/28.16 | **29.31/27.49** | 31.36/29.73 | 38.19/32.13 | 0.87/0.64 | **0.71/0.51** | 1.51/1.13 |
| siMLPe | 24.77/22.23 | **24.10/23.61** | 25.61/24.12 | 29.45/27.67 | **0.72/0.53** | 0.74/0.53 | 1.22/0.93 |
| MCENET | 25.32/23.10 | 25.71/24.36 | **24.32/23.18** | 30.54/29.58 | **0.63/0.51** | 0.71/0.47 | 0.79/0.43 |
| Social-GAN | **20.31/18.29** | 21.21/18.93 | 22.69/17.21 | 27.43/21.35 | 0.59/0.49 | **0.58/1.02** | 0.98/0.62 |
| STGCN | 21.82/20.19 | **20.35/18.72** | 25.12/23.66 | 24.33/20.32 | 0.66/0.37 | **0.54/0.44** | 1.12/0.92 |
| CCVAE | **14.32/12.86** | 17.19/14.25 | 20.15/17.82 | 16.80/12.17 | **0.59/0.57** | 0.75/0.46 | 0.75/0.24 |
| Motionformer | 12.17/10.19 | **11.92/10.88** | 19.11/14.47 | 15.38/13.72 | **0.56/0.44** | 0.68/0.48 | 0.92/0.84 |
| Diffpose | **11.87/10.23** | 13.25/11.18 | 15.87/13.71 | 16.22/13.90 | **0.48/0.41** | 0.69/0.62 | 0.89/0.87 |
| PoseMamba | **12.14/11.33** | 12.92/12.48 | 13.11/11.92 | 15.23/13.92 | **0.55/0.49** | 0.64/0.59 | 0.94/0.90 |

Table 3: Action diversity experiments for the behavioral uncertainty quantification task were conducted using nine baseline models. Left: Model performance on four datasets was assessed using Diversity (↑) metrics **in pixels (px)**. Right: Model performance was evaluated on three datasets using the same metrics **in decimeters (dm)**. Best and second-best results are highlighted in bold and underlined, the gray-shaded ActionRat column denotes the dataset with the highest diversity.

| Method | Penn Action | Animal Kingdom | MARS | ActionRat | Penn Action | Animal Kingdom | ActionRat |
|---|---|---|---|---|---|---|---|
| | Diversity ↑ (px) | | | | Diversity ↑ (dm) | | |
| ABPLSTM | 46.71 | 48.12 | 49.78 | **50.92** | 1.08 | 0.92 | **1.42** |
| siMLPe | 36.71 | 40.31 | 35.12 | **42.19** | 0.87 | 0.91 | **1.16** |
| MCENET | 40.21 | 42.51 | 44.12 | **45.09** | 0.88 | 0.88 | **0.96** |
| Social-GAN | 42.41 | 41.32 | **43.19** | 31.02 | 0.76 | **0.93** | 0.88 |
| STGCN | 35.13 | 31.32 | 34.33 | **39.86** | 0.62 | 0.61 | **0.95** |
| CCVAE | 38.92 | 39.19 | 40.01 | **42.56** | 0.72 | 0.63 | **0.85** |
| Motionformer | 35.89 | **43.10** | 35.68 | 40.53 | 0.67 | 0.65 | **1.05** |
| Diffpose | 37.12 | **43.22** | 36.72 | 41.92 | 0.72 | 0.68 | **0.91** |
| PoseMamba | 35.19 | **41.22** | 39.43 | 40.20 | 0.61 | 0.55 | **0.84** |

ant feeds historical ESI labels together with keypoint sequences into the same model to guide future motion prediction. We compare uncertainty metrics under both conditions in Tables 4 and 5, which show that explicitly incorporating ESI priors effectively reduces predictive uncertainty and improves behavioral prediction performance.

## 4.2 3D ANIMAL POSE ESTIMATION

The ActionRat dataset offers keypoint annotations enabling accurate 3D pose estimation on real data, confirming the reliability of its labels. Furthermore, the synthetic sequences generated by OpenRatEngine exhibit lower 3D keypoint errors in three highly complex limb-occluded actions, demonstrating the high precision of synthetic 3D annotations.

**Experimental details**. We employed DeepLabCut (Mathis et al., 2018) as a baseline to build pose estimation model using ResNet-50 as the backbone, implemented in TensorFlow with the Adam optimizer, a stepwise learning rate decay, a batch size of 8, and 800000 training iterations. Subsequently, 3D pose estimation (Nath et al., 2019) was performed with multi-view sequences and camera calibration parameters. Since the ActionRat dataset uses three cameras but triangulation requires only two, stereo binocular views were specifically adopted to compute 3D keypoints per frame. The real data contained 12 annotated 2D keypoints, but since two ears were not modeled in OpenRatEngine, we excluded them and used the common 10 keypoints for alignment to ensure fair comparison. Following (Patel et al., 2023), 3D keypoints from both DeepLabCut and the synthetic data were reprojected onto the original 2D camera planes for comparison with the ground-truth annotations in ActionRat, and errors were quantified using mean absolute error (MAE).

Table 4: Effect of BCI electrical stimulation instructions (ESI) priors on behavior prediction performance under different stimulation conditions in the ActionRat dataset. Model performance was assessed using MPJPE (↓) and MPJVE (↓) metrics in pixels (px). Best results are highlighted in bold.

| Method | w/ ESI (px) | w/o ESI (px) |
|---|---|---|
| ABPLSTM | **38.19/32.13** | 40.17/34.13 |
| siMLPe | **29.45/27.67** | 34.63/29.12 |
| MCENET | **30.54/29.58** | 33.21/31.86 |
| Social-GAN | **27.43/21.35** | 28.10/22.07 |
| STGCN | 24.33/20.32 | **23.92/19.91** |
| CCVAE | **16.80/12.17** | 17.32/13.36 |
| Motionformer | 15.38/13.72 | **15.17/13.44** |
| Diffpose | **16.22/13.90** | 17.67/14.21 |
| PoseMamba | **15.23/13.92** | 16.92/14.83 |

Table 5: Effect of BCI electrical stimulation instructions (ESI) priors on behavior prediction performance under different stimulation conditions in the ActionRat dataset. Model performance using MPJPE (↓) and MPJVE (↓) metrics in decimeters (dm). Best results are highlighted in bold.

| Method | w/ ESI (dm) | w/o ESI (dm) |
|---|---|---|
| ABPLSTM | **1.51/1.13** | 1.62/1.23 |
| siMLPe | **1.22/0.93** | 1.43/1.13 |
| MCENET | 0.79/0.43 | **0.76/0.45** |
| Social-GAN | **0.98/0.62** | 1.10/0.71 |
| STGCN | **1.12/0.92** | 1.21/0.97 |
| CCVAE | **0.75/0.24** | 0.87/0.38 |
| Motionformer | **0.92/0.84** | 1.03/0.91 |
| Diffpose | **0.89/0.87** | 1.02/0.97 |
| PoseMamba | **0.94/0.90** | 1.12/1.09 |

Table 6: Reprojection errors of 3D keypoints mapped back to the camera 1 image plane for DeepLabCut and OpenRatEngine synthetic data in occlusion actions based on the ActionRat V2 dataset.

| Action | Model | Nose | middleEar | highSpine | Tailroot | Tailmiddle | Tailend | leftforeLeg | rightforeLeg | leftbehLeg | rightbehLeg |
|---|---|---|---|---|---|---|---|---|---|---|---|
| Locomotion | DeepLabCut | **4.01** | **6.16** | 9.71 | **9.39** | 14.94 | 29.94 | 37.91 | **35.82** | 52.20 | 43.66 |
| | OpenRatEngine | 8.32 | 11.27 | **7.89** | 14.24 | **7.84** | **15.12** | **31.03** | 35.89 | **16.79** | **32.05** |
| Sniffing | DeepLabCut | **6.71** | 15.12 | 18.88 | **7.66** | 8.12 | **3.51** | 17.17 | **3.06** | 24.26 | 25.35 |
| | OpenRatEngine | 14.39 | **14.97** | **11.54** | 9.10 | **7.23** | 7.14 | **8.38** | 7.96 | **10.79** | **18.75** |
| Rearing | DeepLabCut | 19.89 | **8.91** | 64.99 | 11.05 | **14.02** | 21.69 | 51.80 | 37.65 | 45.22 | **13.22** |
| | OpenRatEngine | **15.22** | 10.04 | **8.01** | **8.75** | 14.74 | 22.69 | **20.27** | **10.67** | **9.40** | 14.53 |

**Result and analysis**. Table 6 shows 3D pose estimation results for three typical occluded actions in the ActionRat dataset, where locomotion involves drastic posture changes and sniffing and rearing feature severe limb occlusions. Results indicate that DeepLabCut effectively performs 3D pose estimation on ActionRat. However, OpenRatEngine achieves lower errors on most keypoints with smaller reprojection fluctuations. In the rearing action, the "highSpine" error of DeepLabCut reaches 64.99 px, whereas OpenRatEngine is to 8.01 px, benefiting from 3D skeleton construction. Qualitative results in Fig. 3 further show that synthetic sequences maintain strong spatial continuity and stability, overcoming the long-term keypoint loss or deviation often observed in DeepLabCut during occlusion-prone actions such as sniffing and rearing. Results validate the effectiveness of the ActionRat dataset for 3D pose estimation and highlight the advantages of OpenRatEngine-generated synthetic data in improving accuracy and robustness.

## 4.3 SYNTHETIC DATA EVALUATION OF OPENRATENGINE

To verify the effectiveness of synthetic data, we compared its performance with real data in the behavioral uncertainty quantification task and analyzed their similarity. Minimal performance differences demonstrate that synthetic sequences closely match real rats in spatial and temporal features, confirming that OpenRatEngine reduces the domain gap between real and synthetic data (as demonstrated in the supplementary video).

**Experimental details**. We selected six baseline models: PoseMamba, Diffpose, Motionformer, Social-GAN, MCENET, and siMLPe. All were pre-trained on the Animal Kingdom dataset (Ng et al., 2022), where each action segment was processed using a sliding window to extract 20 frames, with the first 10 frames as input and the last 10 as labels. Models were trained for 100 epochs with a batch size of 64. After pretraining, parameters were frozen except for the newly added three-layer MLP. Fine-tuning was conducted using real and synthetic clips from the ActionRat V2 dataset, and models were evaluated by MPJPE and MPJVE metrics under two settings: (i) training and testing on real data, and (ii) training on synthetic data and testing on real data. To measure spatial pose similarity between adjacent frames across action types, we followed (Patel et al., 2023) by projecting synthetic 3D keypoints onto each camera plane and comparing them with ground-truth. For fair comparison as mentioned before, 10 common keypoints (excluding two ears) were aligned

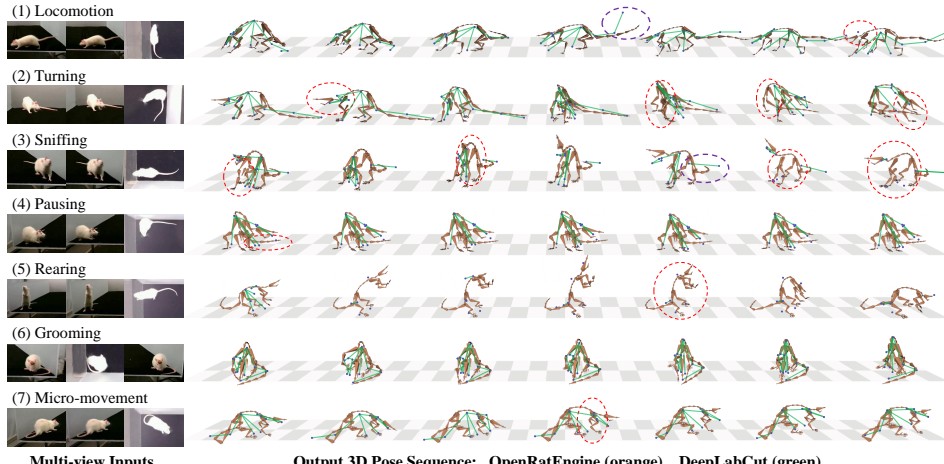

**Multi-view Inputs**          **Output 3D Pose Sequence:   OpenRatEngine (orange) ,  DeepLabCut (green)**

Figure 3: Qualitative comparison between our OpenRatEngine (orange) and DeepLabCut (green). For the seven atomic actions, DeepLabCut exhibits missing keypoints (red) based on triangulation and shows noticeable detection drift (purple) in cases where the tail position varies significantly, while OpenRatEngine effectively avoids the issue of limb occlusion.

Table 7: Comparison of pre-trained models on the ActionRat dataset using MPJPE and MPJVE (px) between real-to-real and synthetic-to-real sequences.

Table 8: Pose variation similarity was evaluated between synthetic (Syn.) and real (Real) actions on adjacent frames based on GW.

| Method | MPJPE (px) ↓ | | MPJVE (px) ↓ | |
|---|---|---|---|---|
| | Real→Real | Syn.→Real | Real→Real | Syn.→Real |
| siMLPe | **36.64** | 39.23 | **30.23** | 31.49 |
| MCENET | **35.78** | 39.44 | 31.22 | **29.97** |
| Social-GAN | **30.12** | 31.58 | **27.39** | 29.66 |
| Motionformer | 28.36 | **27.12** | **27.98** | 30.23 |
| Diffpose | **25.44** | 26.31 | 26.18 | **25.92** |
| PoseMamba | 25.32 | **24.61** | 24.33 | **23.98** |

| Action | Corr. ↑ | Euclidean ↓ | MAE ↓ |
|---|---|---|---|
| Locomotion | **0.8664** | 1.0898 | 0.1601 |
| Turning | 0.8370 | **0.7490** | **0.1222** |
| Sniffing | 0.7713 | 1.4289 | 0.1265 |
| Pausing | 0.7547 | 1.1683 | 0.2456 |
| Rearing | 0.7073 | 1.3202 | 0.2143 |

using the strict quasi Gromov-Wasserstein (GW) distance. Temporal consistency was evaluated based on GW using Pearson correlation, Euclidean distance, and MAE metrics.

**Result and analysis**. First, Table 7 shows minimal performance differences between real-to-real and synthetic-to-real evaluations, confirming the strong consistency of our synthetic data with real distributions and its effectiveness for animal behavior analysis. PoseMamba performs comparably well, indicating a strong ability to model complex feature patterns. Second, the synthetic sequences exhibit high realism. As reported in Table 8, for both static (pausing) and dynamic (locomotion, turning, sniffing, rearing) actions, all Corr. values exceed 0.7, while Euclidean distances and MAE remain low. These results indicate that OpenRatEngine accurately simulates kinematic patterns, avoiding "stiff" or "delayed" motions. OpenRatEngine maintains strong spatiotemporal consistency with real rat motions, effectively reducing the domain gap between real and synthetic data.

## 5  CONCLUSION

We constructed ActionRat, the first multi-view rat behavior dataset integrating real and synthetic data. Building upon common free-exploration behaviors, it introduces a wide range of BCI-based intervention behaviors encompassing both normal and abnormal actions, supporting more behavior understanding tasks. To expand the dataset, we developed OpenRatEngine, a biomechanical model that generates synthetic data with 3D keypoint annotations. OpenRatEngine reduces the domain gap between synthetic and real data. We evaluated ActionRat on behavioral uncertainty quantification and 3D pose estimation tasks, and confirmed the effectiveness of synthetic data.

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

# APPENDIX

## A  OUTLINE

The appendix complements the main content of the paper and provides additional information. We outline the key extended resources.

---

**Overview Supplementary Material**

1. We release the ActionRat dataset, providing a comprehensive and structured description of its composition, collection process, preprocessing steps, applications, distribution, and maintenance. This release aims to advance the research community and encourage broad participation in benchmark evaluations.

2. We will publish the OpenRatEngine virtual rat model, which supports flexible generation of synthetic data, with the goal of facilitating future experiments and methodological development.

3. To ensure transparency and reproducibility, we provide a complete codebase in the supplementary materials, including detailed instructions for training and evaluation protocols.

4. The supplementary materials also include demonstration videos of synthetic actions generated using OpenRatEngine.

---

The remainder of the appendix is organized as follows. Section B provides a detailed description of the ActionRat dataset collection process. Section C presents a comprehensive account of OpenRatEngine. Section D introduces the public datasets used in the experiments. Section E outlines the overall experimental setup, including evaluation metrics, model configurations, and in-depth analyses of additional experimental results. Finally, Section F discusses the current limitations and directions for future work.

## B  DETAILS OF THE ACTIONRAT DATASET

### B.1  BEHAVIOURAL APPARATUS AND DATA ACQUISITION

The open-field served as the primary arena for rat behavior collection, measuring 1 m×1 m and surrounded by 0.5 m high transparent acrylic panels to prevent escape. A top camera was mounted above the arena, and a stereo camera was installed on the side. A diffuse illumination lamp was placed next to the field to minimize shadows and strong reflections, thereby improving video recording quality. Before data collection, six rats were numbered, and their movements under both free exploration and behavioral intervention conditions were recorded sequentially. The arena was sprayed with 70% alcohol before and after each recording to eliminate residual odors. The collection of rat movements was categorized into two types.

**Free Exploration**: Rats were gently placed at the center of the open field. After placement, the experimenter either left the experimental area or remained still, allowing the rats to move freely within the arena. The behaviors collected during this phase reflected their natural activity under no external intervention and were used to assess spontaneous activity, exploration patterns, and anxiety levels. These data also served as a baseline for comparing behavioral changes before and after interventions.

**Behavioral Intervention**: Rats in the open field were remotely controlled through wireless electrical stimulation delivered via a software platform. Exogenous electrical stimulation based on brain–computer interface (BCI) technology enabled precise modulation of locomotion. For implementation, dual-channel stimulating electrodes were implanted in the target brain regions with a reserved interface, connected via wires to a wearable wireless stimulation backpack mounted on the rat's back. The backpack received control commands from the software platform and delivered electrical pulses to achieve remote control. The platform provided a visual interface for parameter

configuration and control operations, while logging all stimulation commands. Stimulation parameters included stimulation frequency, pulse width, pulse number, and stimulus intensity.

## B.2 ANIMAL SUBJECTS

In the ActionRat dataset, we used adult male Sprague–Dawley (SD) rats. Regarding sex selection, most studies in neuroscience and brain–computer interface research prefer male subjects, and extensive knowledge of their physiological and behavioral characteristics has already been accumulated. In contrast, female rats have a short estrous cycle with substantial hormonal fluctuations, which often lead to variability in behavior and physiology and thus reduce experimental consistency. Male rats, on the other hand, exhibit relatively stable testosterone levels and consequently more consistent physiological and behavioral patterns, facilitating reliable data collection and analysis. Nevertheless, in future work, we plan to gradually increase the proportion of female subjects to enhance the diversity of behavioral data.

We initially selected 10 rats, and after a series of training and screening procedures, 6 successfully passed all tests and were used for ActionRat dataset collection. The screening protocol consisted of three tasks—elevated platform, open field, and eight-arm maze—designed to evaluate the rats' responsiveness to electrical stimulation commands. First, rats were placed on an elevated platform, where their innate fear of heights reduced spontaneous movement, allowing a more accurate assessment of locomotor control under turn left, turn right, move forward, and stop commands. Command sequences were delivered via a backpack system controlled by host software, and responses were observed within a 5s window. Rats that met this criterion proceeded to open-field training, where they were guided along predefined paths using stimulation sequences and rewarded with food upon successful completion. Training was repeated until each rat could be reliably controlled for more than 10 consecutive trials, with response times consistently under 5s. Finally, in the eight-arm maze, left-turn or right-turn stimulation sequences were used to direct rats at branching points. Their turning accuracy and decision-making times were recorded, again requiring at least 10 consecutive successful responses within 5s. Each command type was tested in no fewer than 30 trials.

## B.3 CAMERA CALIBRATION

Prior to collecting rat behavior data, we performed time synchronization and spatial calibration of the multi-camera setup to ensure geometric consistency and spatial accuracy of the multi-view data, thereby providing foundation for subsequent 3D pose estimation and behavior analysis. We employed a custom script "get paired image.py" to collect paired calibration images and copied them into the calibration folder generated by DeepLabCut (Nath et al., 2019). A 12×9 black-and-white checkerboard pattern was used as the calibration template, and a total of 80 stereo image pairs were captured using two cameras. The OpenCV library was then used to iteratively calibrate each image pair. During image acquisition, the checkerboard orientation was kept consistent with rotation angles constrained to within 30°, to prevent errors in the order of detected corners. The images were captured at multiple distances and covered diverse regions of the field of view, including the corners and center of each frame. Image pairs with failed corner detection or inconsistent corner ordering between the left and right images were excluded to ensure calibration accuracy. Subsequently, we employed the stereoRectify function of OpenCV for stereo camera calibration to obtain lens distortion parameters as well as intrinsic and extrinsic matrices, and used undistortPoints for image correction. The mean reprojection error for both camera 1 and camera 2 was 0.017 pixels, indicating high calibration accuracy. Following calibration, multi-view videos of candidate rat movements were collected.

## B.4 DATA CURATION

To ensure data quality and applicability, the ActionRat dataset was constructed through rigorous data filtering, standardized annotation, and unified formatting. The dataset is provided in a standardized structure, including raw rat video data from ActionRat V1 (.avi), the ActionRat V2 dataset cropped into clips and organized by action categories (.avi), camera calibration files (.pickle), 2D keypoint annotation files for real data (.csv, .jpg), the virtual rat biomechanical model (.blend), synthetic sequence videos (.mp4), and their corresponding 3D keypoint annotation files (.csv).

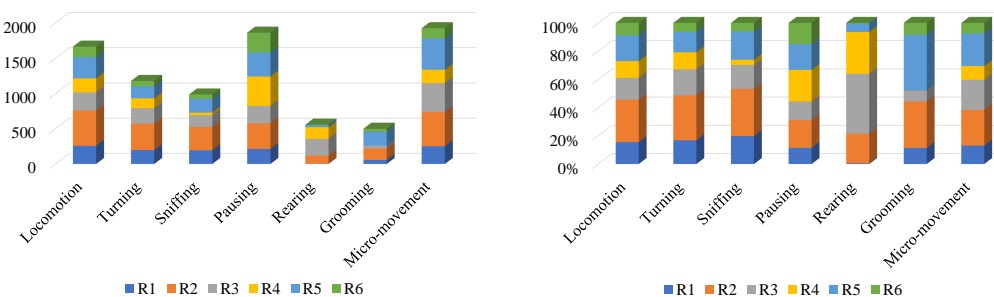

Figure 4: Detailed statistics of the ActionRat V2 dataset presented in absolute counts and percentages, with different colors indicating individual rats.

### B.5 DATA ANNOTATION

**Data Annotation**: For 2D keypoint annotation, a wide variety of frames of actions were considered to train a robust feature detection network. All annotations followed a unified standard, consistently labeling the same anatomical locations. Keypoints that were invisible or occluded were left unannotated. After annotation, the correctness and storage integrity of each body part label were further verified to ensure their effectiveness for training. In addition, the dataset retained a large number of unannotated images, which can be utilized for semi-supervised and self-supervised learning in animal behavior understanding, compensating for the scarcity of existing data resources.

There exists a substantial discrepancy in annotation density between real and synthetic data. The real sequences provide only 12 keypoints, whereas OpenRatEngine can generate 60 3D keypoints. This discrepancy primarily arises from the inherent challenges of annotating small animals such as rats and mice: (i) human visual resolution makes it difficult to accurately distinguish and label fine-grained keypoints; (ii) annotations are often limited by frequent body occlusions; and (iii) large-scale manual keypoint annotation is prohibitively expensive. As a result, dense large-scale annotations are extremely difficult to obtain in practice, a limitation also reflected in existing rodent datasets—for example, TriMouse-161 (Lauer et al., 2022) provides 12 annotated keypoints on mice, Rat7M (Dunn et al., 2021) provides 20 on rats, and Rodent3D Patel et al. (2023) includes only 8. In contrast, OpenRatEngine can automatically provide complete 3D keypoint annotations (60 keypoints) during the synthesis of motion sequences, thereby effectively alleviating keypoint omissions caused by body occlusions and significantly improving the completeness and accuracy of the annotations.

### B.6 DATA STATISTICS

We analyzed the statistical characteristics of the ActionRat V2 dataset to reveal the diversity of action categories. Fig. 4 illustrates the overall behavioral composition, summarizing the sample counts and proportions of seven atomic actions across six rats. The results indicate that locomotion, pausing, and micro-movement occur with higher frequency, whereas rearing and grooming are relatively scarce. Furthermore, Fig. 5 contrasts the behavioral distributions under free exploration and BCI control, highlighting differences in natural habits and locomotor patterns across conditions. During free exploration, micro-movement dominates, reflecting the rats' tendency to remain in corners. Under BCI control, we applied electrical stimulation of varying intensities. Weak stimulation typically resulted in immobility or only micro-movements. As the voltage increased to an appropriate level, rats exhibited movements. At higher voltages, their movements became irregular and uncoordinated. Meanwhile, as shown in Fig. 5(b), exploratory behaviors such as sniffing and rearing were markedly reduced under BCI control. These findings demonstrate that the dataset not only captures the natural behavioral repertoire of rats but also provides insights into how BCI modulation alters their motor patterns. Finally, Fig. 6 presents a detailed rat distribution of actions, laying the groundwork for future analyses of inter-individual variability. In summary, the ActionRat dataset encompasses a wide range of behavioral patterns, providing a solid foundation for fine-grained modeling of animal motion intention.

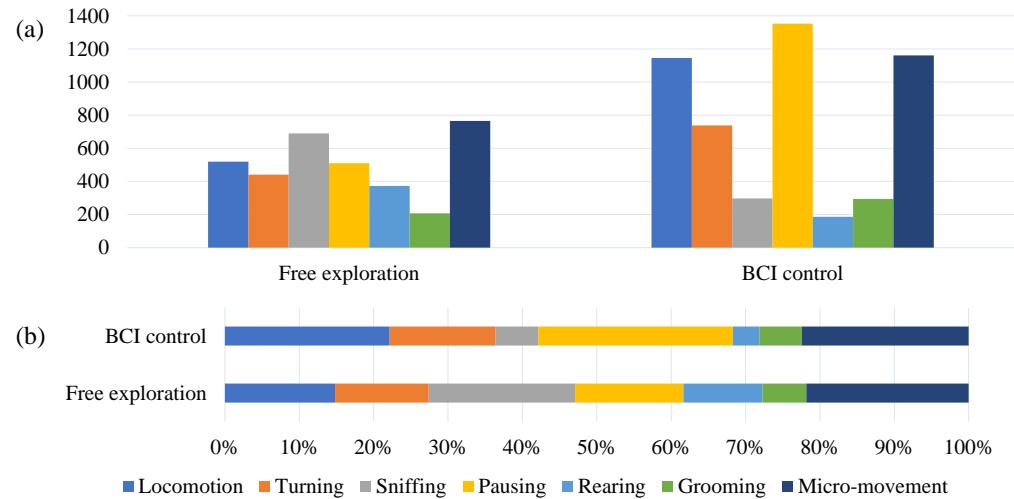

Figure 5: Comparison of rat behavioral distributions under free exploration and brain–computer interface (BCI) control conditions.

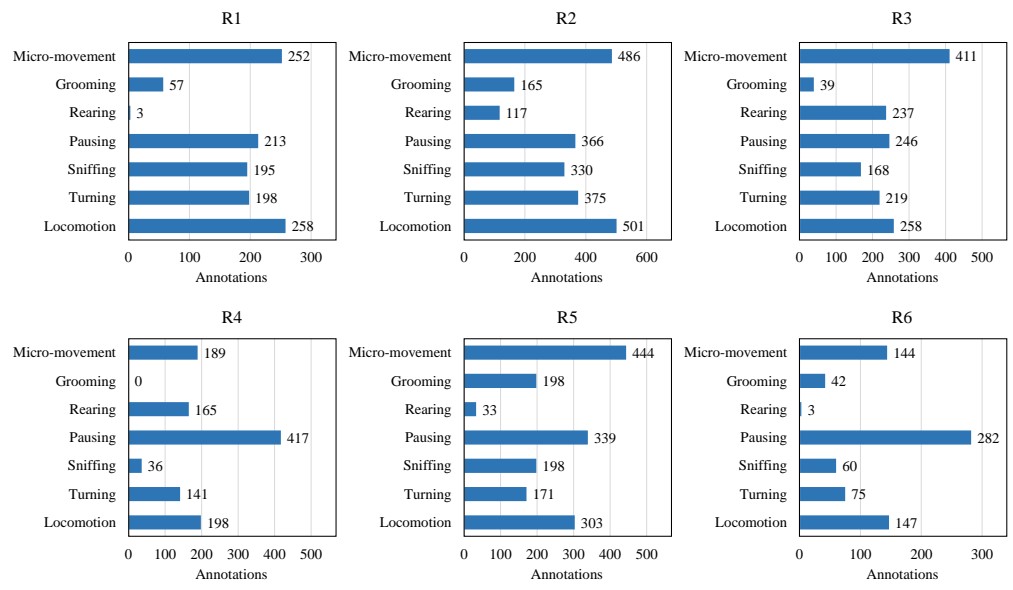

Figure 6: Detailed per-rat distribution of atomic actions across all six subjects.

## C    DETAILS OF BIOMECHANICAL OPENRATENGINE

### C.1    COMPUTED TOMOGRAPHY (CT) SCAN

Since the body length of the rat exceeds the field of view of a single micro-CT scan, we employed a segmented scanning strategy, in which the body was divided into five regions—head, neck, thorax, abdomen, and caudal—and subsequently reconstructed into a complete whole-body model. Each region was subjected to sliced scanning to acquire a large number of tomographic images, as illustrated in Fig. 7. All slices were subsequently registered and stitched using AVIZO software to obtain a complete and continuous three-dimensional skeletal model. CT slices were collected from five rats. Furthermore, to prevent movement during scanning, the rats were anesthetized and immobilized prior to imaging. The employed micro-CT system offers high precision and broad adaptability, with a spatial resolution of 0.5-2 $\mu$m, enabling clear imaging of fine anatomical details. The device

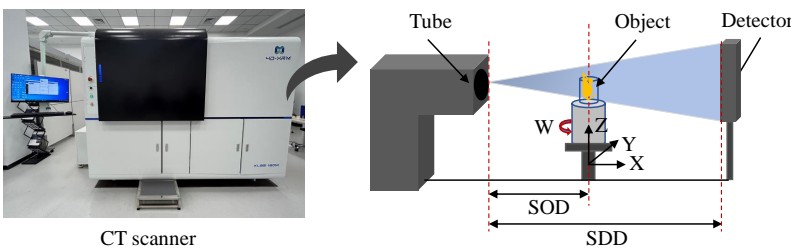

Figure 7: CT scanner.

supports a tube voltage range of 40-160 kV, allowing adjustment of penetration power according to tissue density. During whole-body CT scanning of rats, the specific scanning parameters were set as follows: tube voltage 90 kV, tube current 90 μA, exposure time 0.7 s, and slice thickness 20.308 μm, which is consistent with the voxel size to achieve high-accuracy reconstruction of each layer. These parameters ensured sufficient detail resolution while covering a large field of view, allowing clear reconstruction of the rat's skeletal structure and fine anatomical features. During the construction of the virtual rat, the average dimensions of five rats—measured from body length, spine, limb bones, and tail—were used as scaling references to define the overall structure of the final OpenRatEngine model.

## C.2 SKELETON SIMPLIFICATION

To balance anatomical accuracy with computational efficiency, skeletal simplification is performed in Blender. This process reduces redundant vertices and edges while preserving the overall topology and key structural features relevant to motion modeling, producing a lightweight model. The skeleton was ultimately simplified to 60 bones, as shown in Fig. 8. The detailed articulation relationships among these bones are provided in Fig 9. The simplified structure not only accelerates simulation and rendering but also facilitates the integration of inverse kinematics, rigging, and dynamic control modules within the virtual rat framework.

As shown in Fig. 8, the forelimb is modeled with 3 edges and 3 vertices, where the edges correspond to the scapula, humerus, radius and ulna, and the vertices correspond to the acromioclavicular joint, scapulohumeral joint, and elbow joint. In addition, the forepaw connected to the forelimb is modeled with 5 edges and 9 vertices, where the edges include 1 metacarpal bone and 4 digital bones, and the vertices include the carpal joint, metacarpophalangeal joint, and 4 fingertip joints, which control the movement of the four fingers. The hindlimb is modeled with 4 edges and 4 vertices, where the edges correspond to the ilium, femur, tibia and fibula, and metatarsal, and the vertices correspond to the sacroiliac joint, hip joint, knee joint, and ankle joint. At the same time, the hindtoe connected to the hindlimb is modeled with 6 edges and 11 vertices to control the movement of five toes. The edges correspond to phalanges, and the vertices correspond to the metatarsophalangeal joint, phalangeal joints, and the 5 toe tips. The control of these edges and vertices enables the flexion and extension of the forelimb and hindlimb, as well as the fine movements of the fingers. The spine is modeled with 10 edges and 10 vertices, connecting the cervical vertebrae, thoracic vertebrae, lumbar vertebrae, sacrum, and coccyx. The tail is modeled with 7 edges and 8 vertices, with the last vertex representing the tip of the tail. The head is modeled with 4 edges and 5 vertices, with one vertex each for the nose and mouth. Additionally, the model includes 3 extra control bones with 3 edges and 5 vertices. In the static structure, the rotation angles of all bones are set to zero. The head of each bone was defined as a keypoint, with the coordinate names of all 60 keypoints listed in Table 9. Each bone is equipped with a local coordinate system starting from the initial joint, as shown in Figure 10(b). A custom Python script was developed to automatically extract and export the 3D coordinates of these annotated keypoints in Blender. When the virtual rat OpenRatEngine is driven to generate motion sequences, the corresponding 3D keypoint sequences can be simultaneously exported, thereby enabling automated 3D pose annotation.

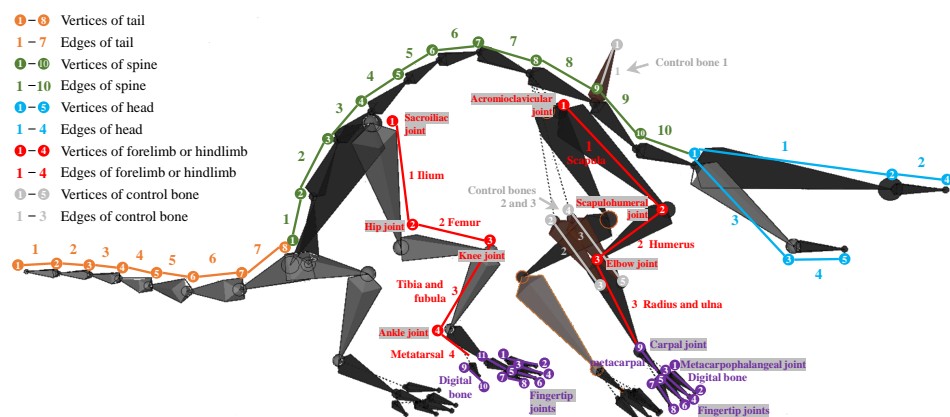

Figure 8: Detailed skeletal linkage representation of the 60-bone virtual rat OpenEngineRat model.

Table 9: Definition of the 60 bones, with the head of each bone selected as a keypoint.

| ID | Definition | ID | Definition | ID | Definition | ID | Definition |
|----|-----------|----|-----------|----|-----------|----|-----------|
| 1 | Spine.005 | 16 | spine.009 | 31 | front_thigh.R | 46 | toe.L.005 |
| 2 | spine.003 | 17 | spine.010 | 32 | front_shin.R | 47 | toe.L.001 |
| 3 | spine.002 | 18 | spine.011 | 33 | front_foot.R | 48 | toe.L.002 |
| 4 | spine.001 | 19 | nose | 34 | front_toe.R | 49 | toe.L.003 |
| 5 | spine | 20 | spine.012 | 35 | front_toe.R.001 | 50 | toe.L.004 |
| 6 | spine.004 | 21 | mouth | 36 | front_toe.R.002 | 51 | pelvis.R |
| 7 | spine.017 | 22 | shoulder.L | 37 | front_toe.R.003 | 52 | thigh.R |
| 8 | spine.018 | 23 | front_thigh.L | 38 | breast.L | 53 | shin.R |
| 9 | spine.016 | 24 | front_shin.L | 39 | breast.R | 54 | foot.R |
| 10 | spine.014 | 25 | front_foot.L | 40 | spine.019 | 55 | toe.R |
| 11 | spine.006 | 26 | front_toe.L | 41 | pelvis.L | 56 | toe.R.005 |
| 12 | spine.013 | 27 | front_toe.L.001 | 42 | thigh.L | 57 | toe.R.001 |
| 13 | spine.007 | 28 | front_toe.L.002 | 43 | shin.L | 58 | toe.R.002 |
| 14 | spine.008 | 29 | front_toe.L.003 | 44 | foot.L | 59 | toe.R.003 |
| 15 | spine.015 | 30 | shoulder.R | 45 | toe.L | 60 | toe.R.004 |

## C.3 BLENDER

Blender is an open-source 3D graphics software that supports a complete workflow, including modeling, rigging, animation, material assignment, and rendering, with flexible configuration of cameras, lighting conditions, and obstacles. We employed Blender as the primary platform to construct the high-fidelity virtual rat model OpenRatEngine and to recreate a realistic open-field experimental environment. The scene was equipped with multi-view cameras and light sources, and materials were defined for Blender's Eevee rendering engine to enhance rendering efficiency and visual realism. During modeling, the skeletal system of OpenRatEngine was rigged under strict anatomical constraints and incorporated custom joint limits to enable biomechanically plausible motion generation. Based on this model, frame-by-frame animations were generated to synthesize behavior data with ground-truth positional labels, seamlessly integrated with Python scripts to achieve automatic 3D keypoint annotation. After completing the action synthesis, the Eevee rendering engine was utilized for efficient rendering, offering a significantly faster performance than Cycles while maintaining high visual quality. Finally, the system simultaneously exports rendered videos and annotated kinematic information for subsequent analysis and model validation.

## C.4 MESH

The mesh of the OpenRatEngine virtual rat exhibits a high level of resolution, with detailed structures shown in Fig. 10. The mesh topology is bound to the nearest functionally relevant skeletal segments derived from the rat's anatomy, with inverse kinematics constraints incorporated to ensure

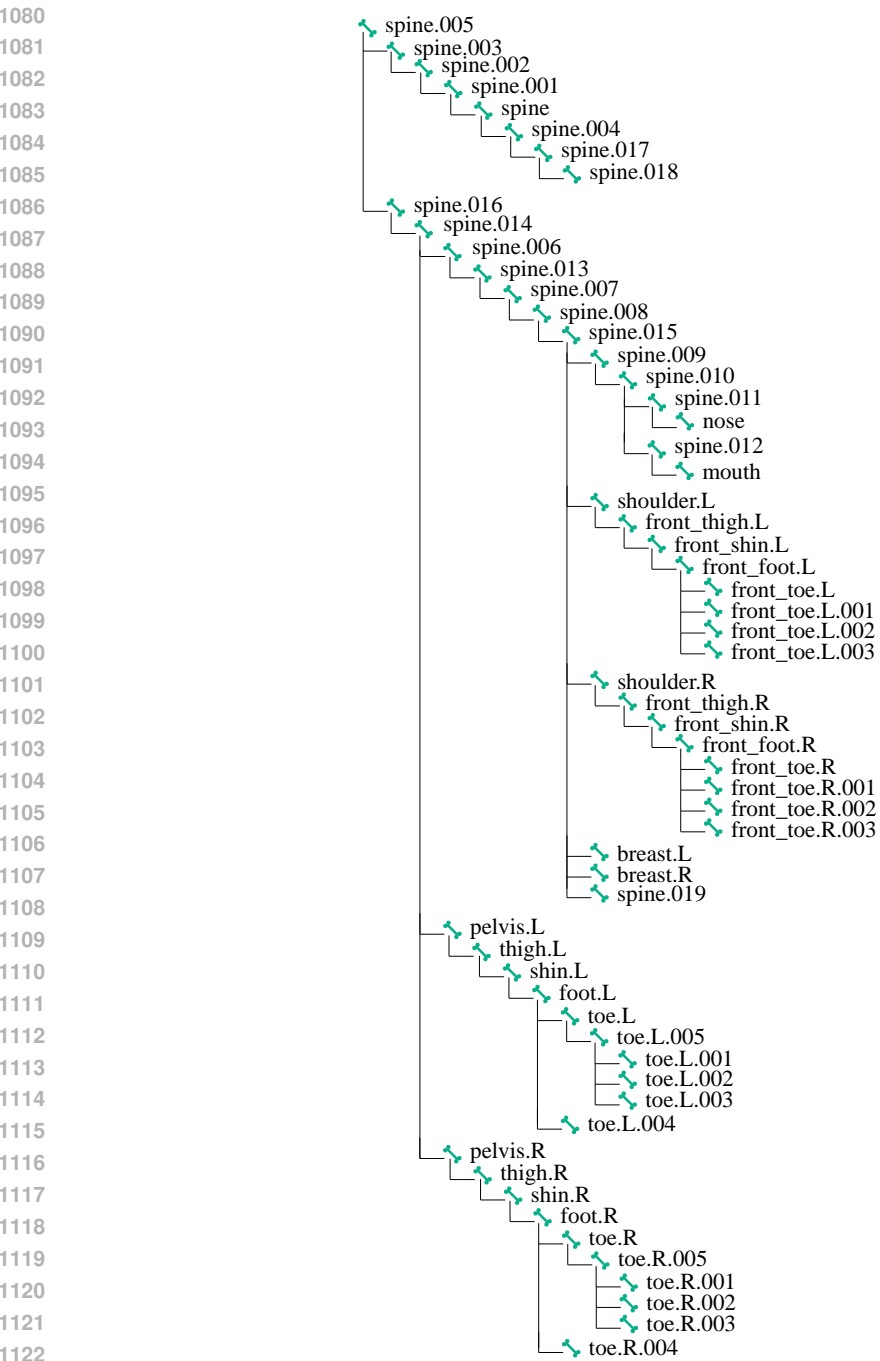

Figure 9: The connection relationship between 60 bones and the bone hierarchy in the Open-RatEngine model.

coordinated motion sequences. The resulting surface geometry displays natural continuity in local regions such as the limbs, tail, and facial contours, while effectively maintaining structural stability and biomechanical plausibility during dynamic movements. The overall appearance is illustrated in Fig. 11.

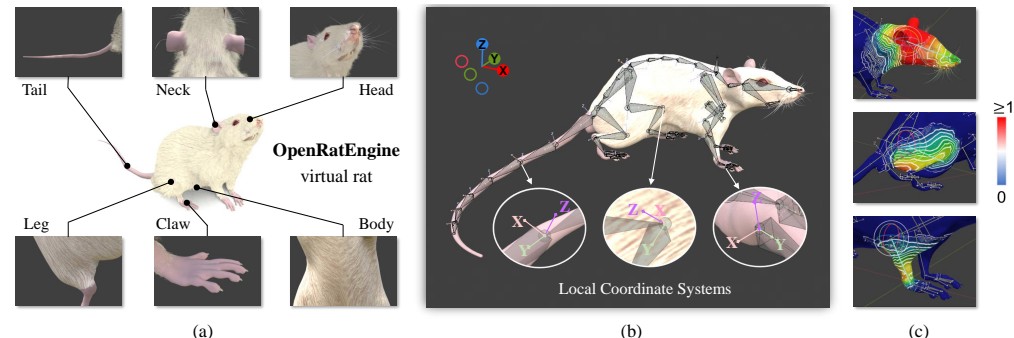

Figure 10: Overview of the OpenRatEngine virtual rat. (a) The virtual rat with overall structure and detailed body parts. (b) Skeletal structure with local coordinate systems, exemplified by those defined at the tail joint, knee joint, and phalangeal joint. (c) Heatmaps of skinning weights for mesh vertices bound to the skeleton.

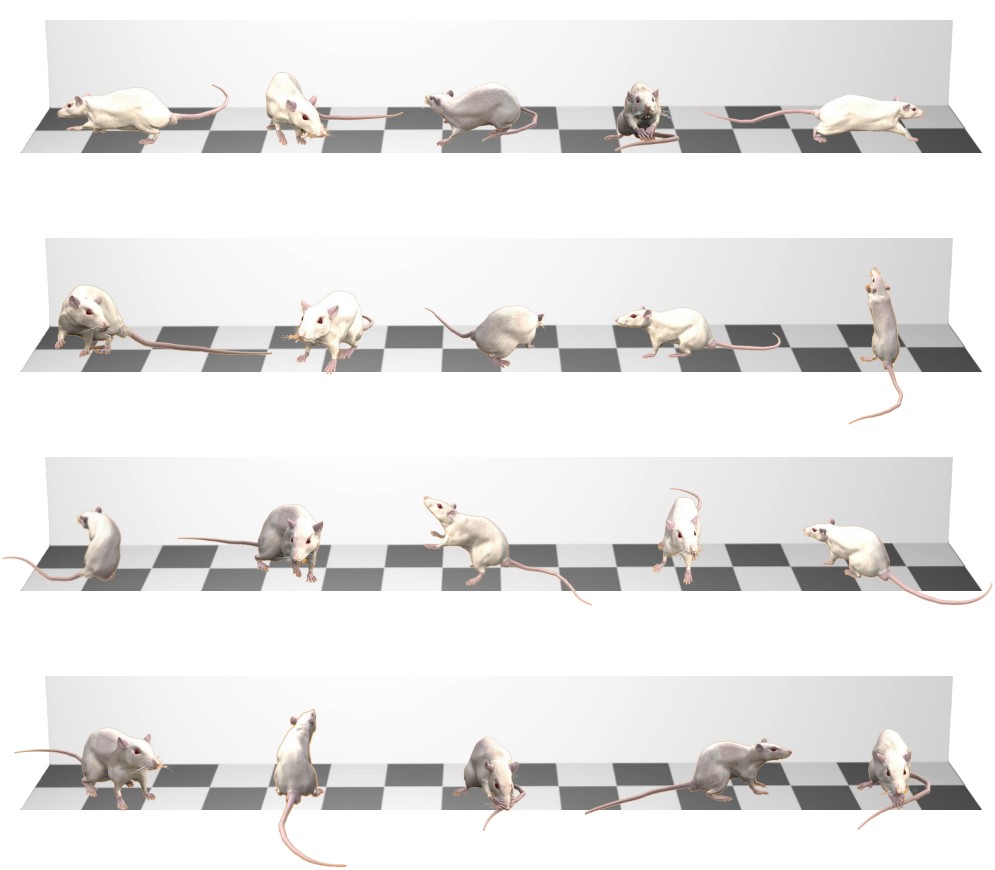

Figure 11: Examples of diverse postures of the virtual rat OpenRatEngine.

## C.5   SYNTHETIC DATA GENERATION

Synthetic data generation is achieved by adjusting the positions and rotations of joints within Open-RatEngine. We first determine the positions and orientations of the virtual cameras to ensure accurate correspondence with target poses observed in the real views. In the three-camera configuration, this requires iterative switching and adaptation across different views until consistent spatial alignment is achieved in each perspective. For the generation of synthetic rat motions, we then es-

tablished a semi-automatic animation pipeline in Blender. Specifically, a set of sparse keyframes is placed along the timeline to represent characteristic poses of the rat, and interpolation is applied between these keyframes to produce smooth transitions, thereby avoiding the laborious process of frame-by-frame adjustments. Through this approach, discrete skeletal postures naturally extend into complete motion sequences, enabling not only efficient data synthesis but also continuity and realism in the generated movements. It is capable of generating structurally complete synthetic data with automatic 3D keypoint annotations.

## D  DETAILS OF THE PUBLIC DATASETS

### D.1  EXPERIMENTAL DATASET

In the main paper, we conduct uncertainty quantification experiments on three public datasets: Animal Kingdom, MARS, and Penn Action. In addition, we supplement experiments in the appendix with results on the Human3.6M dataset to evaluate the differences in uncertainty modeling performance between animals and humans. A detailed description of all four public datasets is provided below.

- **Animal Kingdom**. The Animal Kingdom dataset (Ng et al., 2022) is a large-scale benchmark for research on animal behavior understanding and pose estimation. It provides rich multimodal data, including synchronized RGB videos, depth information, and high-quality 2D and 3D keypoint annotations, covering species-specific behaviors such as locomotion, grooming, feeding, and social interactions. The dataset offers standardized skeletal models and keypoint definitions to support cross-species analysis and additionally includes multi-view video recordings along with fine-grained behavioral labels.

- **MARS**. The MARS (Segalin et al., 2021) dataset is designed for pose estimation and behavioral quantification of freely interacting mouse pairs. Videos were simultaneously recorded with infrared cameras from top and front views, at resolutions of 1024×570 (top) and 1280×500 (front) with a frame rate of 30 fps, yielding approximately 14 hours of footage. From each view, 15000 frames were uniformly sampled and manually annotated by human experts with keypoints. Each mouse was labeled with 7 keypoints in the top view and 11 keypoints in the front view. In addition, three categories of social behaviors—attack, mounting, and close investigation—were annotated frame by frame throughout the dataset.

- **Penn Action**. The Penn Action dataset (Zhang et al., 2013) is a commonly used benchmark for research on human pose estimation and motion modeling in video sequences. It comprises 2,326 video clips covering 15 typical sports and activity categories, including tennis serving, golf swinging, rope skipping, pull-ups, and guitar playing. Each frame is annotated with 26 2D keypoints and includes action bounding box information as well as action category labels, enabling the modeling of both spatial and temporal characteristics of human motion.

- **Human3.6M**. Human3.6M (Ionescu et al., 2013) is one of the largest and most widely used datasets for 3D human motion analysis. It comprises recordings from 11 subjects performing 17 daily activities, including walking, smoking, eating, and phone conversations. Data were captured in a motion capture laboratory using multiple high-resolution cameras and markers to synchronously obtain 3D skeletal data and corresponding 2D image sequences. Consisting of 3.6 million frames with precise 3D joint annotations and camera parameters, Human3.6M provides a rich resource for 3D pose estimation, action recognition, and behavior modeling, and has become a key benchmark in computer vision research.

### D.2  RELATED ANIMAL DATASETS OVERVIEW

Table 1 summarizes representative animal datasets from recent years and compares them with our proposed ActionRat. We provide an overview of these datasets in the following.

- **OpenMonkeyStudio**. The OpenMonkeyStudio (Bala et al., 2020) dataset is a large-scale 3D animal pose dataset captured using 62 synchronized high-resolution cameras. It enables markerless 3D pose estimation, action recognition, and social behavior modeling in freely moving macaques.

- **Rat7M**. The Rat7M (Dunn et al., 2021) dataset a 3D pose dataset comprising approximately 7 million frames of freely behaving Long–Evans rats. Collected from six individuals, each frame is annotated with 20 anatomical keypoints, with synchronized recordings from 12 motion capture cameras and 6 RGB video cameras providing high-precision 3D poses and multiview images. The dataset spans 12 categories of naturalistic behaviors and serves as a benchmark for 3D animal pose estimation task.

- **AP-10K**. AP-10K (Yu et al., 2021) is a dataset for wild mammal pose estimation, comprising 59658 images in total, among which 10015 are annotated with 17 keypoints. The dataset spans 23 mammal families and 54 species.

- **AcinoSet**. AcinoSet (Joska et al., 2021) is a multi-view 3D pose estimation dataset targeting wild cheetahs, capturing complex behaviors such as free running and turning across natural environments. The dataset was collected using six synchronized GoPro cameras, comprising 93 video sequences and a total of 119490 frames. Among these, 7588 frames were manually annotated with 2D keypoints, with 20 keypoints defined per pose. Based on three different 3D reconstruction methods, the dataset further provides 19,915 frames of high-quality 3D pose annotations.

- **Animal Kingdom**. The Animal Kingdom (Ng et al., 2022) dataset comprises real-world videos and images from 850 animal species, spanning six major animal classes. It supports three core tasks. Action recognition task includes 30100 video clips annotated with 140 fine-grained behavior categories. Video grounding task consists of 4301 long video sequences (totaling 50 hours) paired with 18744 natural language descriptions and their corresponding temporal segments. Pose estimation task contains 33099 image frames, each manually annotated with 23 anatomical keypoints across diverse species.

- **TriMouse-161**. TriMouse-161 (Lauer et al., 2022) is a benchmark dataset for multi-mouse 2D pose estimation and individual tracking, comprising 11645 video frames of three unmarked mice freely exploring an open-field arena. Each annotated frame includes 12 keypoints covering anatomical regions such as the head, ears, spine, and tail. The data were collected using a top-view RGB camera at 30Hz with a resolution of 640×480 pixels. A total of 161 frames were manually annotated for model training and evaluation.

- **Animal3D**. The Animal3D (Xu et al., 2023) dataset is specifically designed for 3D pose and shape estimation tasks. It comprises 3379 images collected from the COCO (Lin et al., 2014) and PartImageNet (He et al., 2022) datasets, covering 40 mammalian species. Each image is annotated via a semi-interactive human-in-the-loop process with 26 2D keypoints and segmentation masks. Using these annotations, the SMAL model is fitted to recover the full 3D pose and shape parameters for each animal.

- **Rodent3D**. Rodent3D (Patel et al., 2023) is a 3D behavioral dataset of rodents that provides synchronized thermal infrared, RGB, and depth video streams, specifically designed for behavioral tracking and neuroscience research. It contains approximately 240 minutes of recordings and 4.5 million frames, capturing natural behaviors such as free exploration and foraging within a 1m×1m arena. Data were simultaneously acquired using three thermal cameras and three RGB-D cameras. Approximately 800 frames per thermal view and 200 frames per RGB-D view were manually annotated, with eight 2D keypoints labeled per frame. The dataset provides both 2D annotations and triangulated 3D pose reconstructions.

- **TopViewMouse-5k**. TopViewMouse-5k (Ye et al., 2024) is a top-view image dataset specifically constructed for 2D pose estimation of mice. It comprises approximately 5000 images collected from 13 publicly and privately sourced head-mounted experimental datasets. A total of 322 representative frames were selected using k-means clustering within DeepLabCut, and each frame was manually annotated with 26 keypoints.

- **SyDog-Video**. SyDog-Video (Shooter et al., 2024) is a synthetic dataset designed for dog pose estimation, comprising 500 video clips and a total of 87500 frames. Each frame is annotated with 33 keypoints, 2D bounding boxes, and segmentation masks. The dataset is constructed using the Unity3D engine and incorporates extensive visual and contextual diversity through multi-dimensional domain randomization, including variations in camera parameters, lighting conditions, dog breeds and textures, further enhancing the diversity of the dataset.

- **GenZoo**. The GenZoo dataset (Niewiadomski et al., 2024) is a synthetic 3D animal pose and shape estimation benchmark comprising 1M images, covering hundreds of quadrupedal mammalian species. Each sample is automatically annotated with 3D pose parameters, 3D shape parameters, and a complete mesh, from which a set of uniform-topology keypoints can be derived via the model regressor. All annotations are generated automatically using the parametric SMAL+ model and a conditional generation pipeline, without any manual intervention.

- **CtrlAni3D**. CtrlAni3D (Lyu et al., 2025) is a newly proposed synthetic 3D animal pose dataset covering 10 categories of quadrupeds. It is constructed through a diffusion-based conditional image generation pipeline and consists of approximately 9711 images. Each image provides well-aligned annotations of SMAL parameters (shape $\beta$, pose $\theta$, and camera $\gamma$) and 3D keypoints, along with 2D keypoints obtained by projecting the rendered depth image.

- **MPCH**. The MPCH (Wang et al., 2025) dataset is a benchmark for animal pose estimation that integrates both real and synthetic data. It builds upon three representative existing datasets—AP-10K (Yu et al., 2021), AnimalPose (Cao et al., 2019), and Animal Kingdom–Birds (Ng et al., 2022)—as seed data, and expands them with large-scale synthetic samples generated by the proposed AP-CAP image generation pipeline. For each subset, real and synthetic data are combined at a 1:6 ratio of original to synthetic data, resulting in an overall dataset size of approximately 164K images.

### D.3 Additional Animal Datasets

In addition to the core datasets compared in Table 1, Table 10 supplements and compares a broader set of related animal datasets, thereby underscoring the advantages of ActionRat in task coverage and fine-grained labeling.

- **StanfordExtra**. The StanfordExtra (Biggs et al., 2020) dataset is built upon the 20580 images of the Stanford Dog dataset (Khosla et al., 2011), with additional annotations of 20 keypoints and silhouette masks. After filtering for occlusion and consistency, 8476 valid images across multiple breeds were retained.

- **AnimalWeb**. The AnimalWeb (Khan et al., 2020) dataset is a large-scale hierarchical dataset of 22400 animal faces from 350 species, annotated with 9 keypoints, enabling animal face alignment tasks.

- **PAIR-R24M**. PAIR-R24M (Marshall et al., 2021) is a large-scale and richly annotated dataset for multi-animal 3D pose estimation and behavior recognition. It captures dyadic interactions of 18 distinct pairs of rats, recorded using 12 motion capture cameras and 6 RGB video cameras, yielding approximately 26 hours of synchronized multi-view data and totaling 24.3 million frames. Each rat was equipped with 12 retro-reflective markers placed on the dorsal surface, and 3D keypoint positions were accurately reconstructed via triangulation using an infrared motion capture system. In addition to the 3D pose annotations, the dataset provides frame-wise labels of individual behaviors and inter-animal interaction categories, enabling fine-grained analysis of social behavior and benchmarking of multi-animal tracking

- **Horse-30**. Horse-30 (Mathis et al., 2021) is a high-quality animal pose estimation dataset designed to evaluate out-of-domain robustness. It consists of 8114 frames across 30 individual Thoroughbred horses, with 22 anatomically defined keypoints per frame manually annotated by experts.

- **MouseGPT**. The MouseGPT dataset (Xu et al., 2025) is an open-vocabulary resource for mouse behavior analysis, constructed from synchronized multi-view recordings with eight 4K@60FPS cameras. It spans control and multiple drug-treated groups with 42.8M frames. From 90,059 images, 24 keypoints were manually annotated in 2D and triangulated into 3D keypoints, with 2D bounding boxes added. Furthermore, 270,085 high-quality frame-level open-vocabulary annotations were generated by GPT-4o and refined by human experts, covering dimensions such as Overall, Head, Limb, Torso, Others, and Keywords.

- **MoReMouse-Syn**. MoReMouse-Syn (Zhong et al., 2025) is a high-fidelity dense-view synthetic dataset for mice, generated using a Gaussian Mouse Avatar from the Markerless

Table 10: Comparison of additional animal datasets. "APE": animal pose estimation, "AAR": animal action recognition, "AR": animal reconstruction, "AFA": animal face alignment, "SDG": synthetic data generation, "UQ": uncertainty quantification, "B. Box": bounding box, "Seg. M": segmentation maps, and "SMAL": skinned multi-animal linear model.

| Dataset | Species | Frames | Annot. Method | Annotations | Keypoints | Source | Task |
|---|---|---|---|---|---|---|---|
| StanfordExtra (Biggs et al., 2020) | Dog | 8K | Manual | 2D, Silhouette | 20 | Real | AR |
| AnimalWeb (Khan et al., 2020) | Animals | 22K | Manual | 2D | 9 | Real | AFA |
| PAIR-R24M (Marshall et al., 2021) | Rat | 24.3M | Marker-based | 3D | 12 | Real | APE, AAR |
| Horse-30 (Mathis et al., 2021) | Horse | 8K | Manual | 2D | 22 | Real | APE |
| MouseGPT (Xu et al., 2025) | Mouse | 42.8M | Auto., Manual | 2D, B. Box | 24 | Real | APE, AAR |
| MoReMouse-Syn (Zhong et al., 2025) | Mouse | 18K | Auto. | Mesh | / | Syn. | AR |
| **ActionRat (Ours)** | **Rat** | **609K** | **Auto., Manual** | **2D, 3D, ESI, Cats.** | **12, 60** | **CT, Real, Syn.** | **APE, UQ, SDG** |

Mouse 1 video (18,000 frames at 100 FPS), with 64 rendered views per frame. It automatically provides annotations including RGB images and 3D mesh surface geometry.

# E    DETAILED EXPERIMENTAL SETTING AND RESULTS

## E.1    DETAILS OF METRICS

### E.1.1    MEAN PER JOINT POSITION ERROR (MPJPE)

This metric computes the average Euclidean distance between the predicted and ground-truth positions of all keypoints at each time step within the prediction horizon. It evaluates the model's accuracy in spatial trajectory prediction. Let $N$ denote the number of keypoints, $T_{\text{obs}}$ the length of the observation window, $T_{\text{pred}}$ the length of the prediction horizon, $Y_i^{t+T_{\text{obs}}} \in \mathbb{R}^3$ the ground-truth position of the $i$-th keypoint at frame $t + T_{\text{obs}}$, and $\hat{Y}_i^t \in \mathbb{R}^3$ the predicted position at the $t$-th frame in the prediction window. The MPJPE is given by:

$$\text{MPJPE} = \frac{1}{N \cdot T_{\text{pred}}} \sum_{i=1}^{N} \sum_{t=1}^{T_{\text{pred}}} \left\| Y_i^{t+T_{\text{obs}}} - \hat{Y}_i^t \right\|_2, \tag{6}$$

where $\| \cdot \|_2$ denotes the Euclidean norm.

### E.1.2    MEAN PER JOINT VELOCITY ERROR (MPJVE)

This metric quantifies the average discrepancy in velocity between the predicted and ground-truth trajectories for all keypoints over consecutive time steps, thereby evaluating the model's capability to capture motion dynamics. Let $T$ denote the total number of frames and $N$ the number of keypoints. The velocity of each keypoint is approximated by the first-order finite difference of its consecutive positions. The MPJVE is formulated as:

$$\text{MPJVE} = \frac{1}{T-1} \sum_{t=1}^{T-1} \frac{1}{N} \sum_{i=1}^{N} \left\| \left( Y_i^{t+1} - Y_i^t \right) - \left( \hat{Y}_i^{t+1} - \hat{Y}_i^t \right) \right\|_2. \tag{7}$$

### E.1.3    DIVERSITY

This metric measures the variability among the generated motion sequences, where a higher value reflects a greater diversity of motions. Let $\mathcal{S}_1, \mathcal{S}_2, \ldots, \mathcal{S}_n$ denote the set of $n$ generated sequences, and $\text{MPJPE}(\mathcal{S}_i, \mathcal{S}_j)$ represent the MPJPE between sequences $\mathcal{S}_i$ and $\mathcal{S}_j$. The diversity is defined as:

$$\text{Diversity} = \frac{1}{C(n, 2)} \sum_{i=1}^{n-1} \sum_{j=i+1}^{n} \text{MPJPE}(\mathcal{S}_i, \mathcal{S}_j). \tag{8}$$

### E.1.4    THE QUASI GROMOV–WASSERSTEIN DISTANCE

The quasi Gromov–Wasserstein (GW) distance is invariant to scale changes and rigid transformations and does not rely on the absolute positions of the keypoints. Instead, it represents each pose

using a pairwise distance matrix that encodes the geometric relationships among keypoints, enabling structural alignment across frames. The GW distance is defined as:

$$D_{GW}(\boldsymbol{p}, \boldsymbol{q}) = \|\boldsymbol{M}_p - \boldsymbol{M}_q\|_F, \tag{9}$$

where $\|.\|_F$ denotes the Frobenius norm, and each pose contains $n$ keypoints. The pairwise distance matrix of pose $\boldsymbol{p}$ is denoted by $\boldsymbol{M}_p \in \mathbb{R}^{n \times n}$. The elements $M_{i,j} = \|\boldsymbol{p}_i - \boldsymbol{p}_j\|$ in $\boldsymbol{M}_p$ correspond to the Euclidean distance between the $i$-th and $j$-th keypoints in pose $\boldsymbol{p}$. A smaller $D_{GW}$ indicates a higher level of similarity.

### E.2 DETAILS OF BASELINE MODELS

#### E.2.1 TRAIN-FROM-SCRATCH BASELINES

We conducted comparative analyses of ten baseline models on the proposed ActionRat dataset, with the results presented in Table **??**, 6, and 8. In addition, multiple methods were pretrained, and the corresponding results are reported in Table 7. A detailed review of the baseline models is provided below.

- In PoseMamba (Huang et al., 2025), monocular 3D pose estimation is formulated as a state space model-based framework with linear complexity. At its core lies the bidirectional global–local spatio-temporal Mamba module, where global modeling ensures structural consistency, local geometric reordering enhances fine-grained prediction accuracy, and bidirectional temporal scanning captures cross-frame dependencies, thereby mitigating uncertainties caused by depth ambiguity and local occlusion.

- In Diffpose (Gong et al., 2023), pose prediction is modeled as a reverse diffusion process that progressively transforms an uncertain 3D distribution into a determinate solution, with three key designs introduced. The method initializes the indeterminate 3D pose distribution using 2D keypoint heatmaps and depth statistics, employs a Gaussian mixture model-based forward diffusion to generate intermediate supervisory signals for training, and integrates spatio-temporal context information in the reverse diffusion stage, where graph convolution and attention modules are leveraged to gradually eliminate uncertainty.

- In siMLPe (Guo et al., 2023), the authors leverage the discrete cosine transform (DCT) to represent temporal patterns in the frequency domain, while linear mappings across spatial and temporal dimensions extract informative features. Instead of directly generating future joint coordinates, the model learns the displacement relative to the last observed pose, thereby reducing the complexity of training. Additionally, a velocity-consistency term is incorporated alongside positional loss to better capture motion dynamics.

- In Motionformer (Aksan et al., 2021), the authors introduce an innovative spatio-temporal Transformer framework for 3D motion prediction. This approach explicitly models temporal and spatial dependencies, overcoming the limitations of traditional RNNs that rely on hidden state propagation and frequency encoding. The model embeds joint poses into a high-dimensional space and alternates temporal attention, which captures the historical dynamics of individual joints, with spatial attention, which models inter-joint relationships within the same frame, across multiple stacked layers. This design enables explicit aggregation of historical information during autoregressive prediction, effectively mitigating error accumulation.

- In CCVAE (Cheng et al., 2020a), the authors propose a novel generative modeling framework to address sequence prediction problems characterized by high uncertainty and complex interactions. The core idea lies in jointly encoding historical dynamics, neighborhood interactions, and contextual information, while leveraging a conditional variational autoencoder to learn a latent variable space that captures the diversity of future sequences. The framework employs a dual-encoder architecture to parallelly extract temporal and contextual features, enhancing adaptability to complex environmental constraints through integrated feature modeling.

- In ABPLSTM (Roberts & Segev, 2020), the authors introduce an animal behavior modeling framework that integrates behavior recognition with time-series forecasting. Initially, a Gentle AdaBoost algorithm is employed to learn classifiers from a small set of manually

annotated video frames and then automatically propagate these labels to extensive video data, enabling efficient behavior annotation. The resulting behavioral sequences are subsequently fed into a ABPLSTM network, which is trained to capture long-range temporal dependencies and predict multiple future behavior states.

- In MCENET (Cheng et al., 2020b), the authors encode historical trajectories, interaction patterns, and environmental context into a latent representation to characterize the uncertainty of future motions. Two parallel encoders are employed to extract temporal features from past and future data, while integrating scene semantics, group interactions, and motion dynamics to form a latent distribution that captures complex behavioral dependencies. During inference, multiple samples are drawn from this latent space to produce diverse and socially plausible future trajectories.

- In Social-GAN (Gupta et al., 2018), the authors leverage a generative adversarial framework to model the multimodal nature of future states, enabling the generation of diverse motion sequences that remain consistent with contextual cues. The architecture is built upon an encoder–decoder structure, where LSTM networks capture historical dynamics, and an information aggregation module fuses the states of all entities in the scene to learn interaction-aware representations. The generator samples multiple potential outcomes from a latent space, while the discriminator evaluates their plausibility, driving the model to produce predictions that conform to complex environmental and interaction constraints.

- In STGCN (Yu et al., 2017), the authors address the challenge of modeling and forecasting data with spatiotemporal graph structures. The approach abstracts the data as graph signals, leveraging graph convolutions to capture spatial topological dependencies while employing gated one-dimensional convolutions instead of recurrent networks to efficiently model temporal dynamics. Its core architecture consists of stacked spatiotemporal convolutional modules, where temporal and graph convolutions are alternately combined to achieve cross-domain feature integration.

- DeepLabCut supports both 2D and 3D pose estimation and is applicable to the tracking of single or multiple animals. DeepLabCut 2D (Mathis et al., 2018) achieves high-precision keypoint detection by training on a small number of manually annotated images. Once trained, the model can automatically localize multiple body keypoints in unlabeled video frames and has been widely applied to various species, including rodents, primates, fish, and insects. The primary advantages of 2D DeepLabCut lie in its low requirement for annotated samples, strong generalization capability, and adaptability to diverse behavioral scenarios. Building upon 2D keypoint detection, the 3D extension of DeepLabCut (Nath et al., 2019) incorporates multi-camera views and triangulation to estimate the spatial coordinates of keypoints in three dimensions. DeepLabCut has become an essential tool in the field of animal motion analysis.

### E.2.2 PRETRAIN-FINETUNE BASELINES

The baseline models were pre-trained on the Animal Kingdom dataset (Ng et al., 2022), which encompasses a wide range of quadruped behaviors, and subsequently fine-tuned and tested on a small subset of ActionRat. Through the real-to-real and sim-to-real transfer experiments, the results confirmed that the synthetic data generated by OpenRatEngine can effectively support model training. Models trained on synthetic sequences achieved performance comparable to those trained on real sequences, indicating that the synthetic data is consistent with real data in terms of domain similarity and quality. Furthermore, models pre-trained on Animal Kingdom maintained high performance on rat data in ActionRat, demonstrating their ability to transfer knowledge learned from other species to rat behavior modeling. Therefore, ActionRat not only validates the effectiveness of Sim2Real transfer but also serves as a benchmark for evaluating cross-species knowledge transfer.

### E.3 ADDITIONAL EXPERIMENTAL RESULTS

We trained pose estimation models for each camera view using the 2D annotations from the ActionRat dataset. Based on the multi-view 2D pose estimation results and the provided camera calibration parameters, 3D keypoints were reconstructed. The results in Fig. 12 further validate the accuracy of the 3D keypoints obtained through triangulation, highlighting the precision of camera calibration and the robustness of the multi-view annotation pipeline in the ActionRat dataset.

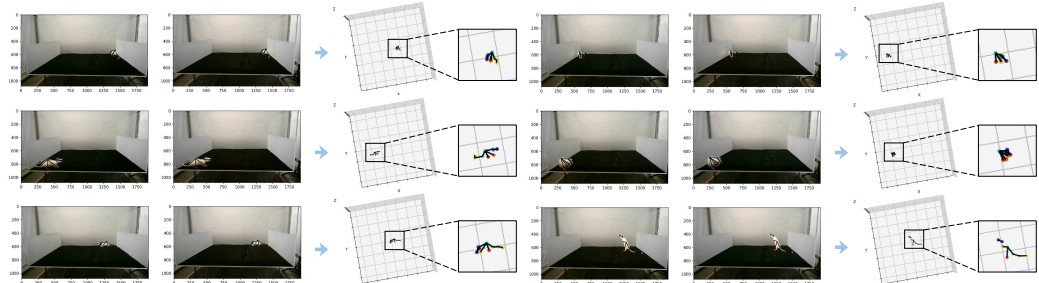

Figure 12: 2D poses captured from stereo views in the ActionRat dataset and their corresponding 3D poses reconstructed via triangulation. Results demonstrate outstanding performance and indicate the high annotation quality of the ActionRat dataset.

Table 11: Comparative experiments for the behavioral uncertainty quantification task were conducted using seven baseline models. Model performance was evaluated on four datasets using MPJPE, MPJVE, and Diversity metrics in decimeters (dm).

| Method | Human3.6M | Penn Action | Animal Kingdom | ActionRat |
|---|---|---|---|---|
| | MPJPE (dm) ↓ / MPJVE (dm) ↓ / Diversity (dm) ↑ | | | |
| ABPLSTM | 1.17/0.99/1.42 | 0.87/0.64/1.08 | 0.71/0.51/0.92 | 1.51/1.13/1.42 |
| siMLPe | 1.08/0.79/1.06 | 0.72/0.53/0.87 | 0.74/0.53/0.91 | 1.22/0.93/1.16 |
| MCENET | 1.15/1.02/0.89 | 0.63/0.51/0.88 | 0.71/0.47/0.88 | 0.79/0.43/0.96 |
| Social-GAN | 0.91/0.69/0.99 | 0.59/0.49/0.76 | 0.58/1.02/0.93 | 0.98/0.62/0.88 |
| STGCN | 0.97/0.77/1.24 | 0.66/0.37/0.62 | **0.54/0.44/0.61** | 1.12/0.92/0.95 |
| CCVAE | 0.98/0.52/1.02 | 0.59/0.57/0.72 | 0.75/0.46/0.63 | **0.75/0.24/0.85** |
| Motionformer | **0.89/0.49/0.82** | **0.56/0.44/0.67** | 0.68/0.48/0.65 | 0.92/0.84/1.05 |

We conducted comparative uncertainty quantification experiments on four datasets across three metrics (in decimeters), as reported in Table 11, including the Human3.6M dataset Ionescu et al. (2013), the Penn Action dataset Zhang et al. (2013), the Animal Kingdom dataset Ng et al. (2022), and our ActionRat dataset. The results further showed that the action diversity predicted using the ActionRat dataset is comparable to that of Human3.6M. CCVAE and MCENET achieved the best and second-best performances on the ActionRat dataset, confirming their strong capacity to model motion uncertainty. On Human3.6M and Penn Action public datasets, Motionformer achieved the highest prediction accuracy, indicating its effectiveness in human motion prediction. However, modeling the uncertainty of rat motion remains highly challenging for existing methods. Rats are strongly dependent on environmental perception, and their locomotor patterns are easily altered by external conditions. This pronounced environment dependence introduces greater uncertainty in motion patterns, making them more difficult to predict than human actions under controlled conditions.

We employed the developed OpenRatEngine to generate a diverse set of synthetic data. As illustrated in Fig. 13, OpenRatEngine demonstrates advanced capabilities in synthesizing fine-grained atomic actions and consistently provides complete 3D keypoint sequences.

# F LIMITATION AND FUTURE WORK

## F.1 LIMITATIONS

For synthetic data generation, we developed OpenRatEngine, a virtual rat modeling framework that leverages Blender's auto-interpolation feature to efficiently generate a large number of motion sequences from a few manually aligned keyframes. Additionally, OpenRatEngine allows the injection of controllable noise into multiple attributes—such as limb displacement, lighting conditions, and camera viewpoints—thereby enhancing the diversity and realism of the synthetic data. Furthermore, with Blender's duplication mechanism, the number of virtual rats can be easily scaled, enabling simulations of multi-rat interactions and social behaviors, which opens new opportunities for group be-

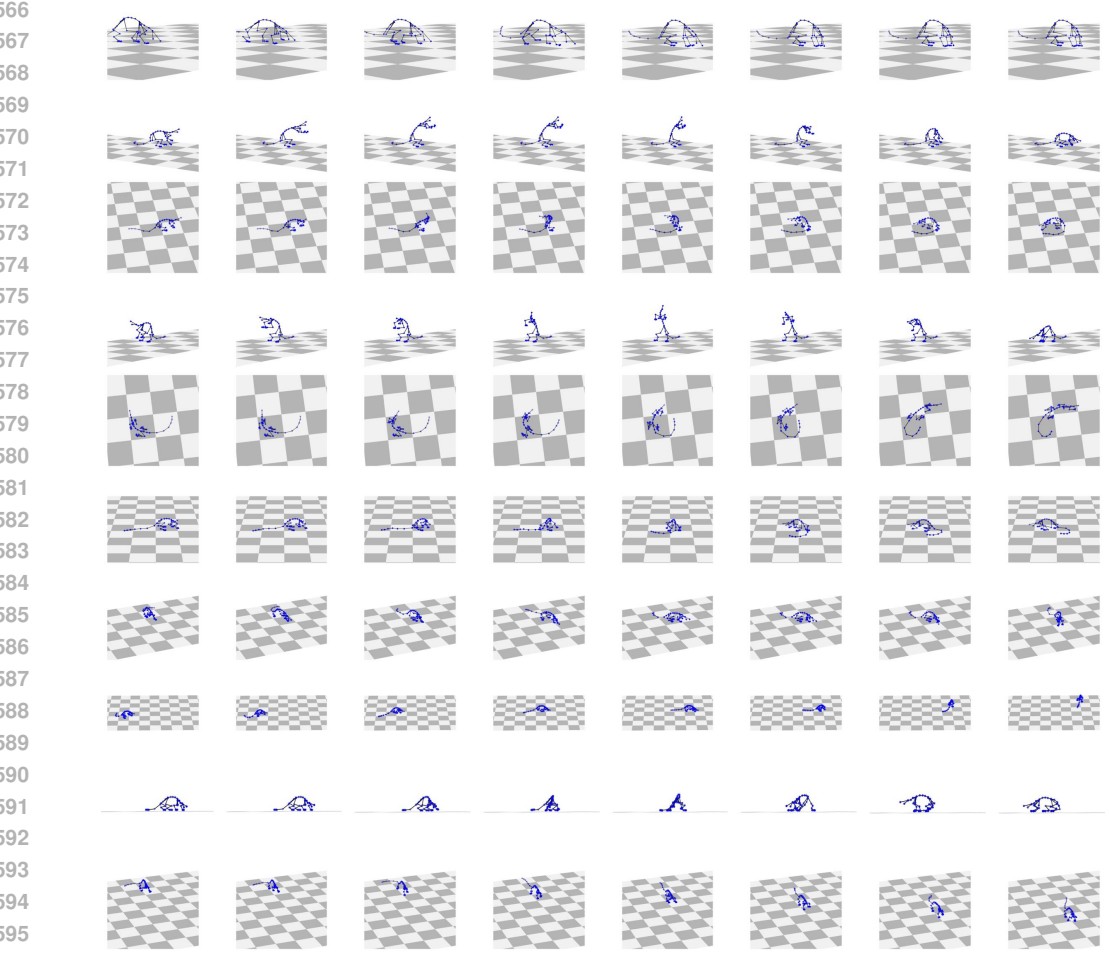

Figure 13: Synthetic motion sequences generated by OpenRatEngine, showcasing fine-grained atomic actions with complete 3D keypoints.

havior modeling research. It is important to emphasize that the use of OpenRatEngine requires users to have a certain level of proficiency in the Blender tool, which may pose a barrier for researchers with limited prior experience.

## F.2 FUTURE WORK

The ActionRat dataset holds significant potential for various research domains, including computer vision and behavioral neuroscience. Specifically, it can serve as a benchmark for tasks such as behavioral uncertainty quantification, 2D/3D pose estimation, action quality assessment, behavior classification, and Sim2Real transfer learning, facilitating both model training and performance evaluation. By supporting these diverse tasks, ActionRat is expected to promote the development of more robust algorithms in animal motion analysis. Moreover, the dataset not only benefits the computer vision research community but also provides an essential resource for scientists engaged in rodent behavioral analysis, fostering cross-disciplinary collaboration. Furthermore, the current data synthesis pipeline adopts a semi-automatic process. In future work, we will advance the mesh fitting stage by developing algorithms that integrate multi-view 2D keypoints, bounding boxes, and contour cues to improve the efficiency and realism of synthetic data generation.

## G    THE USE OF LARGE LANGUAGE MODELS (LLMs)

In this study, we employed large language models (LLMs) to assist with language polishing and expression refinement in order to enhance readability. Importantly, the use of LLMs was strictly limited to linguistic aspects and did not contribute to the design or execution of experiments, nor to the interpretation or analysis of results. All research content was conceived, conducted, and validated solely by the authors, who bear full responsibility for the work.

## H    REPRODUCIBILITY STATEMENT

Reproducibility is a primary objective of this study. To this end, we will release the ActionRat dataset to the research community upon acceptance of the paper, along with the OpenRatEngine virtual rat model. The complete code required to reproduce our experiments will be made available on GitHub, including detailed instructions for environment setup, execution, and adherence to the training and evaluation protocols. These resources ensure that our experimental results can be readily reproduced and extended by other researchers. Further details on data collection, preprocessing, augmentation strategies, and hyperparameter configurations are provided in the appendix.

