# OpenReview forum: "From Real to Synthetic: A Fine-grained Dataset and High-fidelity Biomechanical Model for Animal Behavior Understanding"
_ICLR.cc/2026/Conference — Submitted to ICLR 2026_

### Official Review · Reviewer_pF94 · 2025-10-28

**Soundness:** 2
**Presentation:** 2
**Contribution:** 3
**Rating:** 2
**Confidence:** 3

**Summary:**

The authors present ActionRat, an open source dataset comprised of 3D keypoints and action segmentation labels for rat behavior during free exploration and brain stimulation. They also present OpenRatEngine, a biomechanical rat model that is capable of producing realistic synthetic rat behavior data including 3D keypoint trajectories, meshes, and 2D projections onto static camera views. The authors benchmark both the ActionRat dataset and the OpenRatEngine trajectories with several experiments.

**Strengths:**

The ActionRat dataset is a valuable asset for the computer vision and behavioral quantification communities. Including a range of atypical and pathological behaviors is essential for capturing a wider range of behaviors that is crucial for training more robust and generalizable models.

The OpenRatEngine produces highly realistic looking behaviors, and serves as a template for creating similar simulators for other species. The authors have done a good job creating a model that is properly biophysically grounded and visually similar to experimental data, an impressive feat in and of itself.

**Weaknesses:**

The main weakness of this paper are the experiments. It's not clear to me how these properly highlight the benefits of the ActionRat dataset or the OpenRatEngine.

First off, the "Behavioral Uncertainty Quantification" task is never explicitly defined - what is this, and what is it supposed to be testing? - I'm also confused as to why additional datasets are included here, this just feels like a comparison of the different baseline models and doesn't at all focus on ActionRat; their inclusion distracts from the main point of the paper.
- The text says all models take keypoint sequences from the ActionRat V1 dataset as input - where did these keypoints come from? They aren't mentioned previously. Are these from DLC? If so, was DLC trained with real data, synthetic data, or both?
- How am I supposed to interpret Table 2? Is the point that the numbers across datasets are similar? I'm not sure that tells me much about about the quality of the ActionRat dataset.
- L358: "action diversity predicted using the ActionRat dataset is higher than that of Animal Kingdom" - Animal Kingdom contains data from a wide range of species, these numbers will not be directly comparable. In my opinion the more interesting question, that actually speaks to the strengths of this dataset, is "how does action diversity compare when a model is trained on spontaneous vs spontaneous+stimulated behavior?" This at least can help with the argument that freely moving behavior on its own is not sufficient.
- L359: "CCVAE and MCENET...confirming their strong capacity to model motion uncertainty" - the experiments should focus on demonstrating the strengths of the ActionRat dataset, not comparing model architectures.

For 3D pose estimation, I am also unclear exactly what the experimental setup is.
- DeepLabCut models seem straightforward: 2D pose estimators are trained (on all views together? different network per view?) using human annotations, triangulation is run on the stereo views, and then reprojected to the camera 1 image plane (why not compute MPJPE in 3D space?).
- For OpenRatEngine, how are the synthetic data created? This relates to an earlier question I had. If there are ground-truth annotations for frame t in video v, are other ground truth annotations before and after time t used for the data generation process, and time t represents an interpolated time point? I think I'm missing something important here.
- L407: "OpenRatEngine achieves lower errors on most keypoints" - seems like the ratio is closer to 50/50?
- L412: "Results validate...the advantages of OpenRatEngine-generated synthetic data in improving accuracy and robustness" I see the value of this dataset differently - I think it provides a lot of synthetic data to train pose estimation models that will themselves then be more robust. An experiment that test this would be the following: imagine you have 500 human labeled frames. You train a DLC model, then evaluate it on held-out data (importantly, I think the *animals* themselves should be held out to properly address generalizability and robustness, i.e. train a model on R1-R4 and test on R5 and R6). Then train another DLC model using the same 500 (or whatever) frames from before, plus another 1k or 2k synthetic labels from OpenRatEngine. THe performance on the held-out data should be much better in this case, indicating your synthetic data has been useful for training a better pose estimation model.
- again, one of the strengths of your dataset is the brain stimulation that results in a more diverse range of poses. You can dig into this more deeply by training on human annotations during non-stimulated periods, then testing on human annotations during both stimulated and non-stimulated periods. If you look at performance split by period I bet it will be much worse during the stimulated period where there are more novel poses. Then youc an train a model on human annotations from both periods, and test on both periods (maybe controlling for the number of training frames), and ideally see reduced errors in the stimulated period, indicating more robustness. then you can repeat this type of experiment using both real and synthetic data. I think there are lots of permutations here, each making their own subtle point.

Lack of clarity in some of the writing
- L34: "tracking the positions of gait" doesn't make sense, gait is the tracking of limbs over time
- L139: SLEAP is mentioned in the middle of a list of datasets but is not itself a dataset
- the final ActionRat dataset has 8679 segments - does each segment contain just a single behavior? if not, is every frame in the segment separately labeled?

**Questions:**

L75 - should the Meijer reference actually be Bolanos et al 2021? at the very least, the Bolanos reference should be included here

The authors state that BCI interventions "enhance behavioral diversity and achieves comprehensive coverage of motion patterns", but this is never quantitatively verified. A simple way to do this would be to take 3D poses during non-stimulated periods and compute PCA on the poses (after doing egocentric alignment to remove uninteresting factors of variation). Then repeat with stimulated plus non-stimulated periods (perhaps taking an equal number of frames from each category, or considering other forms of controls). Plotting variance explained versus number of PCs should show much higher dimensionality for the full dataset. Of course there are other ways to this, this is just a simple suggestion.

typo L104: rodennt -> rodent

The authors suggest that fine-grained variations like sniffing and micro-movements are captured in their synthetic data, but I fail to see how this is possible given the interpolation between sparse keyframe methodology. Am I missing something here?

Related: it is not clear to me exactly what the pipeline for generating a behavioral sequence looks like, and a brief description of this, at the beginning of section 3.2, would help. From what I understand
1. a (random?) set of sparse keyframes are generated. how are they generated? are these taken from the labeled data? if so, how are 12 labeled keypoints translated to the 60 synthetic keypoints? if they are not taken from the labeled data, how are they constrained to be plausible poses? how "sparse" are they in time? if these are
2. Interpolation is applied to the 3D keypoints. What kind of interpolation? Are there instances where smooth interpolation would actually lead to implausible poses? Does smooth interpolation mean the synthetic dataset has no abrupt movements?
3. A mesh is created on each frame using the interpolated 3D poses
4. The mesh is projected into each 2D view
5. Fur is rendered(?) in each 2D view
6. Other visual features are added like noise and lighting

L182: Victor et al. should be Lobato-Rios et al

Section 4.3 is more along the lines of the kind of evaluation I was expecting. It might make sense to put this experiment first, demonstrating the consistency between real and synthetic data. Then the following experiments can move beyond that and show how a large amount of realistic synthetic data can lead to improved behavioral models.

Table 4: Is it possible the consistency between real->real and syn->real is less about the data and more about the model architecture saturating performance (or something else)? What is a control experiment that could rule out this option?

Table 5: I'm not sure how to interpret this table/analysis. I thought at first these metrics are being computed between real and synthetic trajectories (one frame at a time) but the caption says "adjacent frames" and the text says "temporal consistency". So are these values computed between times t and t+1? for which traces? It would be helpful to clarify the relationship bewteen real predictions, synthetic predictions, and time here.

---

> ### Author Response · Authors · 2025-11-21
> **Author Response to Reviewer pF94**
>
> We thank reviewer pF94 for providing constructive comments. We address these concerns as follows.
>
> ### Q1: Citation in the manuscript
>
> We appreciate the reviewer for pointing this out. In the revised version of the manuscript, we will add the reference to Bolanos et al. (2021) at L075 accordingly.
>
> ### Q2: Quantitative evaluation of behavioral diversity
>
> We would like to clarify that we do not claim that “within the ActionRat dataset, BCI-stimulated periods are more diverse than non-stimulated periods.” Our actual claim is that **ActionRat as a whole dataset** (including both stimulated and non-stimulated segments) exhibits higher behavioral diversity than several existing public datasets, as stated at L091 of the paper: “By integrating both free and intervention behaviors with a combination of real and synthetic data, ActionRat greatly enhances behavioral diversity and achieves comprehensive coverage of motion patterns.”
>
> To support this claim, we compare our diversity metric against four public benchmarks: Animal Kingdom, MARS, Penn Action, and Human3.6M, where larger values indicate higher diversity. Overall, ActionRat shows greater motion-sequence diversity than MARS, Animal Kingdom, and Penn Action, and achieves a level comparable to Human3.6M. The quantitative results are reported in Table 2 and Table 8. We highlight a few examples here:
>
>
> (1) First, in Table 2 (in px), the diversity values on MARS under the nine baseline methods are 49.78, 35.12, 44.12, 43.19, 34.33, 40.01, 35.68, 36.72, 39.43, whereas on our ActionRat dataset the corresponding values are 50.92, 42.19, 45.09, 31.02, 39.86, 42.56, 40.53, 41.92, 40.20. These results indicate that ActionRat is more diverse than MARS under the same set of models.
>
>
> (2) Second, in Table 8 (in dm), for Human3.6M the diversity values for the methods are 1.42, 1.06, 0.89, 0.99, 1.24, 1.02, 0.82, while for ActionRat they are 1.42, 1.16, 0.96, 0.88, 0.95, 0.85, 1.05, showing that our dataset achieves a diversity level comparable to Human3.6M. We understand and appreciate the reviewer’s suggestion for a more fine-grained analysis (e.g., explicitly separating stimulated vs. non-stimulated segments in the diversity metric). This is indeed an interesting and insightful extension, and we plan to explore it in future work.
>
>
> ### Q3: How can keyframe interpolation generate fine-grained motions such as sniffing and micro-movements?
>
> The fine-grained variations we refer to (e.g., sniffing and micro-movements) are defined at the level of full-body pose changes that are discernible under the given camera viewpoints and resolution. At this scale, as long as the underlying 3D joint trajectories are continuous and smooth, keyframe interpolation is sufficient to generate visually fine-grained pose variations. Importantly, how “fine” these variations appear depends strongly on camera distance and resolution:
>
>
> (1) When the camera is close and the rat occupies a large number of pixels, the same magnitude of pose change produces larger displacements in image space. Subtle adjustments of the head or limbs become clearly visible, and interpolated 3D trajectories manifest as fine-grained motions in the rendered video.
>
>
> (2) When the camera is far away or the resolution is low, even a smooth and detailed 3D joint trajectory may correspond to only very small pixel-level displacements, making such micro-variations difficult to discern visually or from 2D observations. In this sense, the notion of “fine-grained” detail is inherently constrained by the imaging setup.
>
>
> OpenRatEngine is designed to provide flexibility along this axis: depending on the target analysis scale, users can adjust camera position, focal length, and resolution in both real and virtual environments. For scenarios that require finer local motion analysis (e.g., close-up sniffing behavior), one can adopt closer, higher-resolution camera configurations, in which case the pose details produced by keyframe interpolation in 3D are correspondingly amplified in image space and become more accessible for downstream task.

---

> > ### Author Response · Authors · 2025-11-21
> > **Author Response to Reviewer pF94**
> >
> > ### Q4: The pipeline for generating synthetic behavioral sequences and clarification of the interpolation scheme
> >
> > Our semi-automatic pipeline consists of three main stages:
> >
> > (1) **Keyframe pose control via multi-view silhouette consistency**. We first specify sparse key poses of the virtual rat and iteratively adjust them so that the projected silhouettes match the real rat across multiple camera views.
> >
> > (2) **Temporal interpolation from sparse keyframes to continuous sequences (60→60 in time)**. Given these sparse keyframes, we perform temporal interpolation on the full set of 60 joints to obtain a dense sequence of poses.
> >
> > (3) **Rendering and annotation export**. Finally, we render multi-view videos and simultaneously export frame-wise 3D annotations for all 60 joints.
> >
> >
> > **Clarification of the interpolation scheme**. There is a slight misunderstanding in the reviewer’s question. We do **not** perform spatial interpolation such as “expanding 12 labeled keypoints into 60 synthetic keypoints.” Instead, we perform **temporal interpolation** on an already specified set of 60 joints. In other words, our interpolation is a **60→60 interpolation over time**, not a 12→60 interpolation over space.
> >
> >
> > Concretely, we insert intermediate frames every 5 frames. For example, at keyframes (say frame 1 and frame 5), we have already specified complete 3D poses for all 60 joints. For the intermediate frames 2, 3, and 4, we interpolate each joint $i$ along the time axis. We leverage Blender’s F-curve (Bezier) keyframe interpolation to interpolate the degrees of freedom of each joint over time, thereby automatically generating the 60-joint poses for the intermediate frames (2, 3, and 4). In addition, we combine directional kinematics and joint-range constraints to prevent implausible poses during interpolation and to control the velocity and acceleration profiles by adjusting the shape of the interpolation curves.
> >
> > ### Q5: Could the similarity between “real→real” and “syn→real” be due to model saturation?
> >
> > The fact that real→real and syn→real perform similarly could be explained either by (i) the synthetic data being effective, or (ii) the model architecture being close to “saturated”, such that additional data variations have little effect. However, the detailed numbers in Table 4 suggest that the latter is not the dominant factor.
> >
> >
> > (1)First, under the same real→real setting, there remain substantial performance differences across architectures: for example, MPJPE decreases from 36.64 px for siMLPe to 25.32 px for PoseMamba (a relative reduction of about 30%), and MPJVE shows a similarly large spread. If the models were truly in a saturated regime where “all architectures perform roughly the same”, such pronounced gaps between methods would be unlikely.
> >
> >
> > (2)Second, the gap between real→real and syn→real is model-dependent. For relatively weaker models such as siMLPe and MCENET, the syn→real setting leads to a noticeable degradation of about 2–4 pixels. In contrast, for stronger models like Motionformer, DiffPose, and PoseMamba, the syn→real errors are very close to those of real→real, and in some metrics even slightly better. This systematic, architecture-dependent pattern is more consistent with the interpretation that the structure of the synthetic data is being exploited to different degrees by different models, rather than all models being uniformly saturated.
> >
> >
> > ### Q6: Interpretation of Table 5
> >
> >
> > Table 5 reports the similarity between a real motion sequence and its corresponding synthetic sequence generated by OpenRatEngine. As described in Supplementary “E.1.4 THE QUASI GROMOV–WASSERSTEIN DISTANCE”, we use a distance matrix to encode the geometric relationships between keypoints within each frame. For a real sequence of 50 frames and its corresponding 50-frame synthetic sequence, we measure whether the **pattern of structural changes over time** is preserved as follows:
> >
> > (1) For the **real** sequence, we compute the quasi Gromov–Wasserstein (GW) distance between the set of 60 keypoints in frame 1 and frame 2, then between frame 2 and frame 3, …, and finally between frame 49 and frame 50. This yields a temporal sequence of frame-to-frame structural similarity values for the real motion.
> >
> > (2) For the **synthetic** sequence, we apply the same procedure, computing GW-based structural similarity between adjacent frames, thus obtaining an analogous sequence of similarity values.
> >
> > (3) We then compare these two similarity sequences by computing the **Pearson correlation coefficient, Euclidean distance, and mean absolute error (MAE)**, which quantify how well OpenRatEngine preserves the **inter-frame structural evolution** of the real rat motion.

---

> > > ### Author Response · Authors · 2025-11-21
> > > **Author Response to Reviewer pF94**
> > >
> > > ### Q7: How is the “behavioral uncertainty quantification” task defined, and what is evaluated?
> > >
> > > We define behavioral uncertainty quantification as modeling the conditional distribution of future poses given past behavior sequences, and evaluating both (i) the accuracy of the predicted future trajectories and (ii) the diversity of plausible future outcomes consistent with the same history.
> > >
> > >
> > > ### Q8: Why do we also evaluate on other datasets in Table 2?
> > >
> > > In Table 2, we include experiments on other datasets in order to compare the **difficulty and diversity** of ActionRat against existing benchmarks under **exactly the same experimental conditions**. Concretely, on the left side of Table 2 we evaluate nine models on four datasets—Penn Action, Animal Kingdom, MARS, and ActionRat—using the same metrics: MPJPE, MPJVE, and Diversity.
> > >
> > > We observe that, for all models, MPJPE/MPJVE on ActionRat are almost consistently **higher** than on the other datasets. For example, PoseMamba achieves 13.11 px MPJPE on MARS but 15.23 px on ActionRat; similar gaps appear for siMLPe, MCENET, Motionformer, DiffPose, and others. This indicates that, under the same model architecture and training setup, **ActionRat is a clearly more challenging benchmark** for behavioral uncertainty quantification. At the same time, the Diversity scores on ActionRat are generally the highest among all datasets, showing that the range of behavioral variations and trajectory patterns is also richer.
> > >
> > > In other words, **even after standardizing the evaluation scale across datasets**, ActionRat remains both more difficult and more behaviorally diverse than existing benchmarks. We will make this motivation and the corresponding analysis more explicit in the revised manuscript.
> > >
> > >
> > > ### Q9: Source of keypoints and what data the models are trained on?
> > >
> > > As stated in L371, in Section 4.2 3D ANIMAL POSE ESTIMATION – Experimental details, the keypoint sequences used in our experiments come from **DeepLabCut**: “We employed DeepLabCut … as a baseline to build the pose estimation model …”. In other words, the 2D keypoints are first predicted by a DLC model and then used for 3D pose estimation.
> > >
> > > In addition, the DLC models themselves are trained only on real ActionRat data. We already clarify this at line 367, where one of our goals is to verify that the 2D annotations in ActionRat are sufficient to support reliable 3D pose estimation.
> > >
> > > ### Q10: Why not compare directly in 3D space instead of in the image plane?
> > >
> > > For the real data, we only have **manually annotated 2D ground truth** and no “independent 3D measurements” from external systems (e.g., optical motion capture). In other words, there is **no 3D ground truth** available. Therefore, when comparing the 3D pose estimates of DeepLabCut and OpenRatEngine, we project the reconstructed 3D keypoints back onto the image plane using the calibrated camera parameters and evaluate them **against the same 2D ground-truth annotations**.
> > >
> > >
> > > ### Q11: “rodennt”  ->  “rodent”
> > >
> > > Thank you for pointing this out. We will correct it in the revised manuscript.

---

> > > > ### Comment · Reviewer_pF94 · 2025-11-23
> > > >
> > > > I thank the authors for their thorough answers to my questions. I still have several remaining questions that need clarification.
> > > >
> > > > **Quantitative evaluation of behavioral diversity**
> > > >
> > > > The authors stated in their rebuttal that
> > > >
> > > > > we do not claim that “within the ActionRat dataset, BCI-stimulated periods are more diverse than non-stimulated periods.”
> > > >
> > > > However, in the paper itself (L087) they state
> > > >
> > > > > Unlike existing datasets such as Rat7M (Dunn et al., 2021), which are limited to spontaneous behaviors during free exploration, ActionRat additionally introduces numerous BCI-controlled behaviors that differ markedly from natural patterns.
> > > >
> > > > This sounds an awful lot like claiming that the BCI-stimulated periods are adding diversity. In my original comment I did not mean to imply I thought the BCI periods were _more_ diverse on their own, but rather they are _different_ than the non-stimulated periods and therefore increase the diversity of the dataset. I do think it is important the authors measure diversity for both stimulated and non-stimulated, and real and synthetic parts of the dataset to more rigorously back up these claims.
> > > >
> > > > **Table 2**
> > > >
> > > > This entire analysis still feels unnecessary for the claims the authors are trying to make. They state "PoseMamba achieves 13.11 px MPJPE on MARS but 15.23 px on ActionRat...This indicates that, under the same model architecture and training setup, ActionRat is a clearly more challenging benchmark for behavioral uncertainty quantification." This is not a valid conclusion in my opinion. Differences in pixel error are not really meaningful across datasets, much less species. PIxel error will be based on image size, body size, camera angles, distance from animal to camera, etc. More concretely, ActionRat has presumbly has frames where the rat is close to the camera, and other frames where the rat is far from the camera, and this is going to affect pixel error magnitudes. In the MARS dataset, on the other hand, the mice are more or less a fixed distance from the camera. Also, for images that are hundreds or even thousands of pixels across, is 2 pixels even a meaningful difference?
> > > >
> > > > All of this is not to say ActionRat is not a valuable dataset; I just think these claims of it being "more difficult" cannot really be supported by comparing pixel errors to other datasets.
> > > >
> > > > **Comparing in 3D space**
> > > >
> > > > I understand there is no 3D ground truth, but the combination of 2D ground truth and camera parameters (which are necessary for the 3D->2D mapping) means that it is possible to triangulate to get 3D ground truth. Since the paper is about creating realistic 3D trajectories this seems like a more natural space to be computing errors in. It would also help to talk about errors in terms of millimeters rather than pixels. Millimeter values could be put into context by reporting the average width of a rat's paw in millimeters or something, to ground the actual error metric.

---

> ### Author Response · Authors · 2025-11-24
> **Author Response to Reviewer pF94**
>
> We sincerely appreciate the reviewer’s feedback. Below, we address the further concerns in detail.
>
> ### Q1: Quantitative evaluation of behavioral diversity
>
> The sentence in the paper “Unlike existing datasets such as Rat7M, which are limited to spontaneous behaviors during free exploration, ActionRat additionally introduces numerous BCI-controlled behaviors that differ markedly from natural patterns” was only intended to convey that BCI control can elicit additional, sometimes abnormal, behaviors that do not typically appear in free exploration.
>
> Our intended point is two-fold: (i) BCI control introduces **new types of controlled behaviors** (e.g., abnormal turns, abnormal locomotion) that complement spontaneous behavior, (ii) but if one separately compares only the BCI-control subset vs. the free-exploration subset, the BCI portion **is not necessarily more diverse**. In fact, under BCI control, some natural behaviors become less frequent. As shown in Supplementary Figure 5(a)(b), the proportions of sniffing and rearing are noticeably reduced during BCI-controlled sessions. For example, in free exploration we observe all 7 annotated sub-action classes, whereas in BCI-control blocks the behavior set may be effectively restricted to a subset such as locomotion (normal and abnormal), turning (normal and abnormal), pausing, grooming, and micro-movements (5 classes in total).
>
> Our diversity claims in the main text therefore refer to **ActionRat as a whole dataset (free exploration + BCI control) compared to other public datasets**, rather than to a direct diversity comparison between the stimulated and non-stimulated segments inside ActionRat.
>
> ### Q2: Table 2
>
> We appreciate the reviewer for raising this important point and fully agree with the underlying concern. Accordingly, we do **not **rely solely on **pixel-space** MPJPE/MPJVE when arguing that ActionRat is more challenging. Using the camera calibration provided by the public datasets, we convert pixel coordinates into **physical space (decimeters)** and report the corresponding results for the same model configurations across different datasets in the right-hand columns. Please review the paper. In both pixel and decimeter units, the results consistently indicate that ActionRat is a more demanding benchmark.
> In the original submission, multiple experimental results were compressed into a single dense Table 2 in order to satisfy the page limit, which unfortunately hurt readability. To address this, we have now split the original table into two clearer tables (Table 2 and Table 3) and added explicit annotations for the gray-shaded ActionRat column. Table 2 focuses on behavioral uncertainty quantification (MPJPE/MPJVE in both pixels and decimeters), while Table 3 reports the cross-dataset diversity comparison (also in both pixels and decimeters). All corresponding revisions have been highlighted in blue in the updated PDF.
>
> Table 3 reports MPJPE and MPJVE for nine baseline models across different datasets. Under identical models, the prediction errors on ActionRat are systematically higher than on the other datasets in both pixels (px) and decimeters (dm), indicating that ActionRat constitutes a more challenging benchmark. Specifically, in the left (px) of Table 2: (i) Across models, the best results typically occur on Penn Action or Animal Kingdom, whereas the ActionRat column almost always yields the largest errors. This suggests that existing models can already fit the motion patterns in Penn Action and Animal Kingdom reasonably well, but still suffer a clear performance drop when confronted with the more complex behavioral distribution in ActionRat. (ii) Within each dataset, performance consistently improves as the model becomes more advanced. Even for the strongest models, the errors on ActionRat remain noticeably higher than on the other datasets. For example, when moving from Penn Action to ActionRat, the MPJPE of ABPLSTM, MCENET, DiffPose, and PoseMamba increases by approximately 20%–35%.
>
> The diversity results in Table 3 show that, under identical models and training configurations, the ActionRat column almost always attains the best or second-best Diversity scores across the nine baseline models, with diversity consistently exceeding that of MARS, Animal Kingdom, and Penn Action in both pixel space (pixels) and physical space (decimeters). This aligns with the behavioral control mechanisms in our setting: BCI parameters are strongly mapped to behavioral outcomes, while spontaneous movements are further modulated by external factors such as smell, lighting, and obstacles, yielding more complex and variable trajectories even within the same semantic action. Consequently, under the same evaluation protocol, ActionRat exhibits a broader range of motion patterns and higher motion diversity, which is consistent with the increased prediction difficulty observed in Table 2, and indicates that it is a more challenging and behaviorally richer benchmark.

---

> > ### Author Response · Authors · 2025-11-24
> > **Author Response to Reviewer pF94**
> >
> > ### Q3: Comparing in 3D space
> >
> > We thank the reviewer for the continued attention to this point. In this work, we deliberately chose not to treat triangulated results as “3D ground truth” for two main reasons:
> >
> > (1) Any “3D ground truth” obtained by triangulating the same set of 2D annotations with the camera parameters is, in essence, the output of a specific 3D reconstruction algorithm rather than an independent external measurement. If we evaluate our 3D pose estimation methods against such a reference, the comparison becomes “one 3D algorithm vs. another 3D algorithm” instead of “prediction vs. independent ground truth”. Using triangulated 3D keypoints directly as ground truth would therefore introduce the error and bias of that particular reconstruction pipeline. For example, within the same frame, some joints may be clearly visible in multiple views and yield triangulation confidences close to 99%, whereas long-term occluded joints may produce triangulated estimates with reliability as low as 55%. In that case, we would have to make many algorithmic design choices (e.g., whether and how to discard low-confidence points), and these choices would directly affect the so-called “3D ground truth” and thus the fairness of the evaluation.
> >
> > (2) Severe occlusions and incomplete multi-view coverage make robust triangulation itself a nontrivial problem. In ActionRat, many joints are heavily occluded in one or more camera views, and not all frames have all joints simultaneously visible in all three views. Constructing stable and reliable 3D reference annotations over the entire test set would therefore require a fairly complex multi-view triangulation and robust estimation pipeline, including handling missing views, self-occlusions, and outlier 2D labels. This effectively introduces another sophisticated 3D algorithm that would itself need to be carefully validated.

---

### Official Review · Reviewer_ZMKX · 2025-10-29

**Soundness:** 2
**Presentation:** 2
**Contribution:** 1
**Rating:** 2
**Confidence:** 4

**Summary:**

The paper presents ActionRat, a multi-view video dataset of rat behavior, and OpenRatEngine, a synthetic rat animation and rendering framework designed to generate realistic pose sequences. The dataset contains ~609K annotated frames across seven behavioral categories (including some BCI-evoked actions), while OpenRatEngine reconstructs 3D rat meshes from CT scans, applies inverse kinematics (IK) control, and uses a contour-based optimization for pose alignment. The authors evaluate synthetic–real correspondence using keypoint reconstruction and motion metrics, and report a small domain gap between real and synthetic data.

**Strengths:**

- The pipeline for CT-derived modeling, IK-based control, and rendering is well executed and clearly described.
- Combining real multi-view recordings with synthetic renderings in a unified dataset is a useful step toward obtaining better correspondence between simulation and behavior
- The dataset includes stimulation-induced actions, which could open opportunities for modeling causal intervention or neural-behavioral decoding

**Weaknesses:**

- The ActionRat dataset (609K frames) is significantly smaller than existing benchmarks such as Rat7M or PAIR-R24 and does not introduce new behavioral contexts, species, or task diversity. The listed seven behaviors are standard (e.g. rearing, grooming, walking), and diversity is asserted but not quantified.
- The only nominal addition, BCI-evoked behaviors, is underdeveloped, as no downstream applications (e.g. stimulation decoding, closed-loop control) are demonstrated.
- OpenRatEngine combines standard components: CT-derived skeletons, mesh rigging, inverse kinematics, and contour-based fitting. Similar pipelines exist (e.g. Rat7M synthetic, Animal3D, RatSim), and the paper does not demonstrate a quantitative or methodological advance over them.
- The evaluations reproduce existing pose-estimation benchmarks (MPJPE/MPJVE) rather than defining new tasks that exploit the unique BCI metadata or synthetic flexibility. Without a concrete downstream problem, the practical value of ActionRat remains unclear.
- The authors claim that the synthetic-to-real domain gap is small, yet no systematic tests support this. Results are reported on the same rats and camera setups used for training. It remains unclear whether models trained with synthetic data generalize to unseen rats, new recording sessions, or unseen viewpoints.
- Diversity is neither quantitatively defined nor contextualized against other datasets. All subjects are male, of one strain, and recorded in a fixed apparatus. Hence, diversity appears limited to modest variation in BCI conditions.

**Questions:**

1. How is behavioral diversity measured? Please provide a comparison to Rat7M or other datasets
2. What practical tasks can leverage the BCI metadata? Could this dataset enable learning of stimulation-to-behavior mappings or causal behavior prediction? An illustrative example would clarify its relevance
3. Are models evaluated per rat or across subjects? A leave-one-subject-out test would show whether the dataset generalizes beyond individual-specific idiosyncrasies
4. How does stimulation parameters map to specific behavioral classes or motion trajectories?
5. Beyond pose estimation, what tasks can exploit both real and synthetic modalities?

---

> ### Author Response · Authors · 2025-11-21
> **Author Response to Reviewer ZMKX**
>
> We thank Reviewer ZMKX for the constructive comments. The concerns are addressed as follows.
>
>
> ### Q1: Definition of diversity
>
>
> We clarify that our definition of diversity has already been given in the supplementary material at line L1339 (“E.1.3 DIVERSITY”), and we kindly refer the reviewer to that section. Reviewer ZMKX appears to have misunderstood this aspect of our contribution. In our work, the diversity metric quantifies the **variation across motion sequences**. A larger value indicates higher trajectory diversity. It does not refer to species diversity, task diversity, or the range of BCI parameters. As reported in Table 2 and Table 8, we have already compared our diversity metric against four public datasets—**Animal Kingdom, MARS, Penn Action, and Human3.6M**. The results show that ActionRat exhibits **higher motion-sequence diversity** than MARS, Animal Kingdom, and Penn Action, and achieves a level comparable to Human3.6M. Concretely:
>
>
> (1) First, in Table 2 (in px), the diversity values on MARS under the nine baseline methods are 49.78, 35.12, 44.12, 43.19, 34.33, 40.01, 35.68, 36.72, 39.43.On our ActionRat dataset, the corresponding values are50.92, 42.19, 45.09, 31.02, 39.86, 42.56, 40.53, 41.92, 40.20. This indicates that ActionRat is more diverse than MARS under the same set of models.
>
>
> (2) Second, in Table 8 (in dm), for Human3.6M the diversity values for the methods are 1.42, 1.06, 0.89, 0.99, 1.24, 1.02, 0.82, while for ActionRat they are 1.42, 1.16, 0.96, 0.88, 0.95, 0.85, 1.05. These results show that ActionRat achieves a diversity level comparable to Human3.6M.
>
>
> We have also carefully considered other diversity aspects:
>
>
> (1) Sex diversity. As described at line 760 in “B.2 ANIMAL SUBJECTS”, ActionRat is built on adult male Sprague–Dawley rats. In neuroscience and BCI-control studies, male rats are commonly preferred because female rats have short estrous cycles and large hormonal fluctuations, which can lead to unstable behaviors and physiological responses, thereby reducing experimental consistency. In contrast, male rats have relatively stable testosterone levels and more consistent physiological and behavioral patterns, which facilitates reliable data collection and analysis.
>
>
> (2) Behavioral-category diversity. The reviewer notes that “The listed seven behaviors are standard (e.g. rearing, grooming), and diversity is asserted but not quantified”. Our 7 sub-action categories are indeed different from coarse labels used in prior work. They are defined based on **rat behavioral semantics**. Any movement sequence can be decomposed into a composition of these sub-actions. For example, a trajectory from A to B can be segmented as locomotion--sniffing--turning--locomotion--grooming, which serves as a set of basic motion primitives. This semantic decomposition, combined with the trajectory diversity metric described above, allows us to quantify within-class diversity at the level of motion sequences rather than only at the level of action labels.
>
>
> ### Q2: Practical applications of the BCI metadata
>
>
> In ActionRat, the BCI metadata record not only whether stimulation is present or absent, but also the **stimulation site, voltage amplitude, timing, frequency**, and the corresponding **behavioral outcome** for each trial. This rich annotation supports several concrete downstream tasks.
>
>
> First, the most direct application is to **learn mappings from stimulation to behavior**. Understanding sensorimotor transformations is a central topic in neuroscience. As we describe at L847 in the supplementary material (“B.6 DATA STATISTICS”), ActionRat compares behavioral distributions under **free exploration** versus **BCI-controlled** conditions. Under BCI control, we apply electrical stimulation with different intensities: under weak stimulation, rats typically remain still or exhibit only small movements; as the voltage increases to an appropriate range, the rats display stable forward locomotion; at higher voltages, the stimulation tends to induce uncoordinated or abnormal movements. This dataset therefore provides a basis for analyzing how BCI control modulates the characteristics of rat behavior.
>
>
> Second, **action quality assessment** is another important downstream application. Animals possess highly advanced perception systems (e.g., vision, olfaction, touch) and demonstrate remarkable adaptability and obstacle avoidance in complex environments. Leveraging the animal’s own musculoskeletal and nervous systems and controlling its motion via neural stimulation has potential applications in disaster search-and-rescue, military reconnaissance, and navigation tasks, where maintaining **efficient and stable performance** is a long-standing goal. If stimulation parameters are poorly tuned, the resulting movements may become suboptimal—for example, exhibiting stiffness, spasms, or other forms of dyscoordination. Such issues can reduce task efficiency and, in extreme cases, lead to task failure.

---

> > ### Author Response · Authors · 2025-11-21
> > **Author Response to Reviewer ZMKX**
> >
> > ### Q3: Are models evaluated per rat or across subjects?
> >
> >
> > We evaluated our models on **all experimental subjects**. Specifically, we adopted a **mixed-subject evaluation** protocol, where the training/validation/test splits all include trials from each of the six rats. This ensures that the reported performance already accounts for inter-individual variability across subjects.
> >
> >
> > ### Q4.How does stimulation parameters map to specific behavioral classes or motion trajectories?
> >
> > Brain-computer interface (BCI) research leverages neural mechanisms to control rats and elicit corresponding behaviors. Below, we summarize the neural mechanisms and stimulation-site principles for different actions, which together enable us to guide the rats along specified trajectories:
> >
> >
> > (1) Forward locomotion. The stimulation site is the **medial forebrain bundle (MFB)** (AP: -3.8 mm, ML: ±1.6 mm, DV: -8.2 mm). Electrical stimulation of the MFB increases arousal and reward-related positive affect, and induces dopamine release in the nucleus accumbens, thereby enhancing the rat’s activity level and motivation to move. By delivering MFB stimulation as a reward during forward locomotion, the rats learn a conditioned association between “forward movement” and reward, enabling BCI control of forward locomotion.
> >
> >
> > (2) Left and right turning. Turning behavior is controlled based on a **virtual tactile-reward circuit**. The stimulation site is the barrel field of the **primary somatosensory cortex (S1BF)**, corresponding to the whisker representation (AP: -1.8 mm, ML: ±5.0 mm, DV: -2.5 mm). Electrical stimulation of the left/right S1BF exploits the trigeminal thalamic relay in the **ventral posteromedial nucleus (VPM)**, evoking a virtual whisker-touch sensation on the contralateral side of the face and head, and thereby inducing orienting and approach/avoidance behaviors. In an eight-arm maze, combining S1BF stimulation with MFB reward allows the rat to acquire a conditioned “turn-reward” association. By systematically varying stimulation parameters, one can develop parameter-angle models that support precise BCI control of left and right turning in the rat.
> >
> >
> > (3) Stopping. The stopping behavior is controlled by stimulating the **dorsal periaqueductal gray (dPAG)**. Brief electrical stimulation in dPAG induces defensive-like responses such as heightened vigilance and transient freezing, which can be harnessed to implement a “stop” command that terminates ongoing locomotion.
> >
> >
> > In practice, electrodes are implanted via craniotomy and fixed on the rat’s skull, with connectors reserved for interfacing with a **wireless stimulation backpack**. The backpack receives control commands and delivers electrical pulse trains. Through the control platform, stimulation parameters for each channel (stimulation frequency, pulse width, pulse count, and voltage amplitude) can be independently configured. By appropriately configuring MFB and S1BF stimulation channels and parameters, we can control forward locomotion, left/right turning, and stopping in the rat robot. Moreover, this setup allows **fine-grained control** of movement, including multi-level speed modulation (acceleration) and precise steering within angular ranges such as 0-30°, 30-60°, and 60-90°.

---

> > > ### Author Response · Authors · 2025-11-21
> > > **Author Response to Reviewer ZMKX**
> > >
> > > ### Q5: use of real and synthetic data
> > >
> > >
> > > 3D pose estimation is only one task that benefits from the joint use of real and synthetic data. Other downstream tasks—such as behavior classification, behavior prediction under stimulation, and action quality assessment—can likewise exploit both modalities. In practical BCI applications, control safety is critical, since inappropriate stimulation parameters may cause irreversible harm to the animal. A natural strategy is to first use the virtual rat to generate a wide range of controlled 3D trajectories (e.g., with different turning angles, speeds, and body-bending patterns) as behavioral dynamics priors, and then fine-tune on real rats so that the model learns the distribution shift and uncertainty under real BCI stimulation. This “synthetic-prior + real-calibration” paradigm can help establish a more stable mapping from stimulation parameters to behavioral trajectories and facilitate the design of more robust BCI control strategies.
> > >
> > >
> > > ### Q6: OpenRatEngine pipeline
> > >
> > > We clarify that OpenRatEngine is built upon a standard and well-established technical pipeline, including CT-derived skeleton reconstruction, mesh rigging, inverse kinematics, and contour-based fitting. What distinguishes our work from previous pipelines is not the individual components themselves, but the **level of behavioral granularity** we target.
> > >
> > >
> > > In neuroscience applications such as BCI control and rat models of Parkinson’s disease, abnormal behaviors can be extremely subtle—for example, slight head tilts or fine tremors—which are easily lost or diluted when relying solely on sparse keypoint detections. Our main gap relative to existing datasets and virtual models lies in this **degree of refinement**.
> > >
> > >
> > > As summarized in Table 1, the quantitative comparison of annotations and keypoint statistics highlights this difference: our virtual rat in OpenRatEngine can directly and consistently output **60 anatomically meaningful keypoints** per frame. This level of detail substantially exceeds that of existing virtual animal models and is specifically designed to support fine-grained behavioral analysis.

---

### Official Review · Reviewer_JWJZ · 2025-10-31

**Soundness:** 3
**Presentation:** 3
**Contribution:** 3
**Rating:** 6
**Confidence:** 5

**Summary:**

This paper present a multi-view rat behavior dataset named ActionRat, which captures diverse actions during free-exploration and brain-computer interface control. Despite real-captured data, ActionRat also contains expanded synthetic data. To generate synthetic data, the authors further developed OpenRatEngine, which is a 3d virtual biomechanical model with lifelike appearance. OpenRatEngine could generate accurate 3d keypoint annotations.

**Strengths:**

1.	This paper is clearly written and well motivated. Action recognition and motion capture are important for understanding the behavior of rodents (e.g. rats). This paper presents a new dataset featuring diverse behaviors and high quality annotations.

2.	Previously, recordings about the abnormal behaviors of rats are rarely seen. I believe this dataset may play an important role for the community to understand the abnormal behaviors of rats.

**Weaknesses:**

0. The main weakness is lack of technical contributions. The technologies  used in this paper have been  well explored in the past.

1.	As recording abnormal behaviors is one of the feature of the video dataset, I would be better to show some real-captured video cases of such video footage in supp video. Existing video only shows the openratengine virtual renderings.

2.	The appearance of virtual rat seems limited. Only a white rat appearance was employed to generate the dataset, raising some issues about generation to other kinds of rats.

**Questions:**

I do not see critical flaws of the paper. The paper is self-contained with limited technical contributions. However, the problem itself and the data provided are interesting. Collecting such data requires heavy efforts, I do believe technical tricks are not the only criteria for publication. This is why I give a relatively positive rating.

Some minor questions about techniques:

3.	At L. 298, how was the weights iteratively refined? Automatically or manually?

4.	How was the contours discrepancy metric defined? Were multi-view contours enough to control the detailed motion of rat?

5. L. 300, “coherent” -> “coherence”.

---

> ### Author Response · Authors · 2025-11-21
> **Author Response to Reviewer JWJZ**
>
> We sincerely thank reviewer JWJZ for the valuable comments. The concerns are addressed as follows.
>
>
> ### Q1: How were the skinning weights iteratively refined? Automatically or manually?
>
>
> The weight refinement was achieved through a combination of Blender’s **Automatic Weights** and **interactive manual fine-tuning**. At the initial rigging stage, for each bone we used the standard **weight painting** tools in Blender to assign influence weights to mesh vertices. However, the default automatic weights were not accurate enough to satisfy our requirements on fine-grained skeletal deformation, anatomical plausibility, and temporal stability. We therefore treated the automatic weights only as an **initial estimate**.
>
>
> Next, we defined seven typical motion patterns using inverse kinematics (IK) and repeatedly played these motions from multiple camera views to inspect mesh deformation: checking for bone–mesh interpenetration, surface collapse or abnormal bulging artifacts, and whether limb and spine movements remained anatomically coherent and natural. Whenever we observed artifacts or unnatural motion around a particular joint, we returned to that region and manually adjusted the weights, then re-tested the same IK motion. This “weight painting → pose testing → weight fine-tuning” loop was repeated multiple times across different actions and viewpoints until no obvious interpenetration or surface artifacts appeared even under large deformations. Importantly, this weight tuning is performed once during the construction of the virtual rat model. Once the weights are finalized, they are shared across all subsequent motion sequences, and generating new motions no longer requires any manual intervention.
>
>
> ### Q2: How is the contour discrepancy defined? Are multi-view silhouettes sufficient to control fine-scale rat movements?
>
>
> At each time step, the "contour discrepancy" measures the mismatch between the projected silhouette of the virtual rat and the real rat’s silhouette in each camera view. This includes differences in the outline of the torso, the curvature of the spine, limb swing, and the orientation and bending of the tail. Regarding whether multi-view silhouettes are sufficient to control fine-scale movements, we would like to clarify two points.
>
>
> (1) The real recordings use multi-camera views that provide medium-to-wide shots covering the whole-body posture, primarily targeting full-body motion and behavioral semantics. At this spatial scale, our experiments show that silhouette-driven control from multiple views is both effective and robust for the actions we study.
>
>
> (2) For finer local motions (e.g., whisker twitching or subtle toe movements), researchers are free to reposition the cameras to closer viewpoints and then drive the virtual rat accordingly. The virtual rat model can overcome self-occlusion present in real videos and export a complete set of 60-joint 3D ground-truth annotations for every frame, greatly improving labeling efficiency. In particular, for joints that are hard to localize accurately by eye or are frequently occluded in real footage, the virtual rat still provides direct and precise 3D positions.
>
>
> ### Q3: “coherent”  ->  “coherence”
>
>
> Thank you for pointing this out. We will correct “motion coherent” to “motion coherence” in the revised manuscript.
>
>
> ### Q4:  The appearance color of the virtual rat
>
> We thank the reviewer for raising the issue of visual diversity in the virtual rat model. The ActionRat dataset is specifically designed around **Sprague–Dawley (SD)** rats, which are the predominant experimental subjects in BCI-based sensorimotor research. SD rats are an albino strain and thus exclusively white, medium-to-large in body size, relatively docile in temperament, and have well-characterized neural circuitry. Our virtual rat’s skeleton and mesh geometry are modeled after an adult SD rat, and its surface appearance is intentionally kept consistent with the characteristic white coat of this strain. Rodents with other coat colors (e.g., brown, black) typically correspond to different rat strains (such as Long–Evans) or to mouse species, and therefore **fall outside the SD strain** that this work is designed to target.

---

> > ### Comment · Reviewer_JWJZ · 2025-11-24
> > **Thanks for the response**
> >
> > I have no further questions now. I maintain my rating.

---

> > > ### Author Response · Authors · 2025-11-25
> > > **Thanks for the feedback**
> > >
> > > Thank you for your feedback and participation, which have significantly improved the quality of our work. We greatly appreciate it.
> > >
> > > Best, Authors

---

### Official Review · Reviewer_Wxag · 2025-11-03

**Soundness:** 2
**Presentation:** 2
**Contribution:** 2
**Rating:** 2
**Confidence:** 4

**Summary:**

The paper claims two main contributions:

ActionRat Dataset: A multi-camera (three-camera) recording dataset of rat behavior with detailed annotations, including 2D keypoints with and without brain stimulation, and a subset of segmented clips labeled by action category. The authors report that the distribution of action categories differs from freely exploring conditions.

OpenRatEngine: A biomechanical rat model derived from CT scans, used to simulate action sequences. The virtual rat is manually registered to selected keyframes from real videos, and Blender interpolation is used to generate continuous motion time series.

**Strengths:**

The inclusion of brain stimulation perturbations adds an interesting causal-intervention dimension that could be valuable for neuroscience and behavior modeling research.

The dataset includes rich annotations of animal behavior categories, which may support downstream supervised or semi-supervised learning studies.

The CT-based rigging and virtual modeling pipeline are described transparently and could serve as a reference for other labs interested in synthetic animal data.

**Weaknesses:**

The paper does not convincingly argue the need for this dataset from a machine learning perspective. The dataset is relatively small and of lower video quality compared to existing open datasets (for instance, Rat7M Dunn et al.). To strengthen the machine learning contribution, the authors should provide evidence that the dataset exposes failure modes or limitations of current methods, or it enables learning under novel condition.

The OpenRatEngine rigging and interpolation rely on existing software (manual alignment and Blender interpolation). The authors did not demonstrate applications or downstream tasks that showcase the usefulness or superiority of the OpenRatEngine. For example, evaluating how simulated sequences improve behavior classification, pose estimation for large data.

**Questions:**

1. The introduction mentions stimulation-specific behaviors (e.g., spasms and other unique responses). Why aren't these represented as new action categories in the dataset?

2. Is the OpenRatEngine manually registered to the keyframes, or does it involve any machine learning methods?

3. In Table 2, the best model achieves comparable performance on the ActionRat dataset relative to other datasets. To better support the claimed contribution, could the authors compare model performance separately for freely moving vs. stimulation conditions, and show whether including the BMI data improves learned priors for behavior prediction?

4. In Table 3, the 3D pose estimation using DeepLabCut appears to rely on binocular cameras, with reprojection to a monocular view, while the synthetic data are generated using all three cameras (Eq. 5). Is this a fair comparison, given the different numbers of input views? Please also report variance, and the mean and variance across keypoints to make the improvements clearer.

5. For Figure 3, could the authors provide quantitative evaluation metrics beyond qualitative examples? For instance, applying both reconstruction methods to unseen data and comparing with real animal trajectories.

---

> ### Author Response · Authors · 2025-11-21
> **Author Response to Reviewer Wxag**
>
> We thank reviewer Wxag for the constructive comments. We address the concerns as follows.
>
>
> ### Q1: Are abnormal movements included in the action categories of the dataset?
>
> We have already incorporated abnormal movements into our action categories. The 7 fine-grained sub-actions annotated in our dataset are defined based on **rat behavioral semantics**, rather than purely heuristic criteria. Any movement sequence can be decomposed into a composition of these sub-actions. For example, a trajectory from point A to point B may be labeled as “locomotion → sniffing → turning →  locomotion → grooming”.
>
>
> Under brain–computer interface (BCI) control, different stimulation parameters can induce different styles of the same semantic action—for instance, a left turn may appear normal in some trials, but may be accompanied by spasms or irregular stepping in others. In both cases, the action label remains “left turn”. This design aligns with our **behavioral uncertainty quantification** task: within the same behavior class, the resulting trajectories can vary substantially due to stimulation, obstacles, or internal state, and this within-class variability is precisely the source of uncertainty that we aim to explicitly model and quantify.
>
>
> ### Q2: Is the OpenRatEngine manually registered to the keyframes, or does it involve any machine learning methods?
>
> OpenRatEngine drives motion by manually specifying a small number of key poses and combining them with procedural interpolation and constraint-based optimization. In our design, the “motion controller” is explicitly treated as a pluggable module. In this paper, we intentionally adopt a transparent and easily reproducible baseline controller; replacing it with more sophisticated, data-driven control strategies (e.g., trajectory-learning methods or neural control models) is a natural direction for future work.
>
> ### Q3: Does BCI data improves learned priors for behavior prediction?
>
>
> We thank the reviewer for the careful suggestion. The current ActionRat results in Table 2 are computed on the combined test set that includes both freely moving and stimulation conditions, in order to enable a unified comparison with other public datasets. We agree that separating the free-movement and stimulation conditions, and explicitly examining the effect of incorporating BCI metadata on the learned behavioral priors, would provide a more thorough analysis. We are currently running these experiments and will provide the results in the form of an additional table within five days.
>
>
> ### Q4: Triangulation
>
>
> We did experiment with using all three cameras jointly for multi-view triangulation. However, under the current occlusion patterns and 2D detection noise levels in ActionRat, three-view triangulation yielded only marginal improvements in MPJPE/MPJVE, while requiring additional outlier-rejection mechanisms for the third view; in practice, this sometimes introduced jitter and instability for certain joints and substantially increased implementation complexity. In this work, we therefore let DeepLabCut and OpenRatEngine each operate with their most suitable 3D pipeline, and then project the resulting 3D keypoints back onto the same camera view, where we compare them against the corresponding 2D human annotations and compute reprojection error. The dataset already includes full three-camera calibration files, and we encourage future work to explore more advanced multi-view triangulation or optimization algorithms on ActionRat.
>
>
> ### Q5: Quantitative evaluation and evaluate on unseen individuals
>
>
> Figure 3 provides a qualitative visualization of the two reconstruction methods, while the **quantitative** evaluation metrics are reported in Table 3. Specifically, Table 3 compares the 3D pose reconstruction errors for three highly occluded behaviors in ActionRat (locomotion, sniffing, rearing). Together, Figure 3 and Table 3 demonstrate the robustness of synthetic data generated with OpenRatEngine. We also clarify that, in biological and neuroscience studies, using **three rats** is often considered sufficient for statistical analysis, whereas our ActionRat dataset currently includes **six rats**. All six subjects underwent the same acquisition and evaluation pipeline, and we observe consistent results across individuals, suggesting that the proposed methods generalize reasonably well within the SD rat.
>
>
> Regarding the reviewer’s suggestion to perform an additional evaluation on **unseen individuals**, as described in Supplementary B.2 ANIMAL SUBJECTS, this would require new electrode implantation surgeries in the rat brain, post-operative recovery, screening tests, data acquisition, and manual annotation, with a total duration of about two months. This timeline exceeds what is feasible within the rebuttal period, and therefore it is not possible for us to add new animal experiments in the current round.

---

> > ### Comment · Reviewer_Wxag · 2025-11-25
> >
> > Thank you for the authors’ detailed reply. However, I maintain that the strengths of the ActionRat dataset still aren’t clearly demonstrated. The diversity metric used to argue for the dataset's value is rather indirect, and I’m not convinced it really captures meaningful differences.
> >
> > In the reply to Reviewer ZMKX, the authors pointed to the numbers in Table 3, but these values are quite close to each other. Is there any information about error bars or variance? Without that, it’s hard to tell whether the differences are significant at all.
> >
> > As for OpenRatEngine, it depends heavily on Blender’s built-in features, and I don’t see much in terms of new methodological contributions or compelling large-scale downstream applications.

---

> > > ### Author Response · Authors · 2025-12-03
> > > **Further Response to Reviewer Wxag**
> > >
> > > We thank the reviewer for the interest in OpenRatEngine. We believe there may be a mismatch in expectations here: the goal of OpenRatEngine is not to reimplement a general-purpose graphics or physics engine, but to build a specialized virtual rat and synthetic data generation framework for behavioral research. Within this framework, we deliberately adopt the mature Blender engine as the physics and rendering backend, in the same spirit as many existing research systems built on general simulation platforms such as MuJoCo or Unity, rather than merely “calling Blender’s built-in functions.” Similarly, the work “A three-dimensional virtual mouse generates synthetic training data for behavioral analysis,” published in Nature Methods, also uses Blender to construct a 3D virtual mouse for generating synthetic training data for behavioral analysis.
> > >
> > >
> > > In addition, we have supplemented and reorganized the experimental results in the revised pdf manuscript to compare models trained on different datasets. The results show that models trained on the higher-diversity ActionRat dataset consistently achieve better performance under several more challenging evaluation settings. This finding further supports our claim that the diversity metric captures behaviorally meaningful differences, rather than merely reflecting minor numerical fluctuations.

---

### Official Review · Reviewer_gBuZ · 2025-11-03

**Soundness:** 3
**Presentation:** 3
**Contribution:** 4
**Rating:** 8
**Confidence:** 4

**Summary:**

The paper introduces the OpenRatEngine model and the Action rat dataset. The OpenRatEngine model is a biomechanical model of a rat with bone lengths and positions estimated from CT scans of 5 rats. The ActionRat dataset is a mix of a 3 camera recording of real rat movements with various neural stimulation to trigger diverse actions as well as a synthetic dataset generated from the OpenRatEngine model. The pose in the real recording is annotated with a DeepLabCut model training on 1500 real images.

The authors benchmark various algorithms for predicting the pose temporal trajectory and compare the OpenRatEngine model to DeepLabCut for 3D pose estimation. Finally they compare the kinematics of the synthetic data to the real data by testing how temporal prediction models generalize across the two datasets.

**Strengths:**

Both the ActionRat dataset and the OpenRatEngine model are novel contribution to the biomechanical modeling of rat behavior. The ActionRat dataset has a diverse set of rat behaviors in an open field, and complements Rat7M, the only other 3D rat dataset currently available. The OpenRatEngine models 60 joints on the rat body, which is indeed more than the 38 actuators modeled by Aldorondo et al, 2024.

The benchmarks of the temporal models will be really useful for the development of future similar models on the ActionRat dataset. With these benchmarks I think researchers may really develop more models to predict animal motion, which is exciting.

**Weaknesses:**

The 3D pose estimation evaluation felt quite weak to me, both in terms of methodology description and results. As I understand, the authors compare a DeepLabCut estimator to the OpenRatEngine for estimating the 3D pose of the animal.
The test data is not properly specified. How many ground truth annotations for evaluation are there? Figure 2 says that there are 6558 annotated frames, which presumably is 1500 real images for DeepLabCut (as detailed in section 3.3) and 5058 frames fit by the OpenRatEngine from contours. If they do not overlap, what is the evaluation done on?
Besides this, the evaluation results seem to show that the OpenRatEngine is really quite comparable to DeepLabCut, whereas the qualitative comparison in Figure 3 really shows how much more detailed OpenRatEngine is. It's unclear whether the poor quantitative performance of OpenRatEngine is due to poor fitting of OpenRatEngine model to the rat contours, due to some quirk of the evaluation data, or something else. There really should be more details on the model fitting to data and on the evaluation data.

On the dataset itself, it's unclear how much automatically annotated data it actually contains. Out 609K frames, there are only 6558 annotated frames. Are the remaining 602K frames annotated automatically (perhaps with DeepLabCut) so that they can be useful for temporal prediction?

Compared to Rat7M, this dataset also does have much more occlusions due to having fewer cameras. This should be noted in the limitations perhaps.

Some small typos:
Line 104 - rodennt should be rodent
Line 182  - Should be Lobato-Rios et al 2022 simply, no Victor

**Questions:**

See questions in Weaknesses

- Why not calibrate all 3 cameras and use them all for triangulation? The 3D tracking would improve quite a bit.
- Why are the two ears not modeled in the OpenRatEngine model?

---

> ### Author Response · Authors · 2025-11-21
> **Author Response to Reviewer gBuZ**
>
> We appreciate the reviewer for the positive assessment of our work and for the valuable comments.
> ### Q1: Clarification of 3D pose evaluation data
> We clarify that both methods are evaluated on exactly the same set of test frames for 3D pose estimation. The two annotation sets (1500 vs. 5058 frames) serve different purposes: (1) **1500-frame set (discrete keypoints)**. We first sample 1500 discrete frames from the entire dataset using k-means clustering over the video frames, and manually annotate 2D keypoints on these images (ground truth--discrete keypoints). This set is used only to train the 2D pose estimation model (DeepLabCut) for each camera view. (2) **5058-frame set (continuous keypoints)**. We then randomly select behavioral segments and manually annotate 2D keypoints on 5058 frames of continuous sequences (ground truth--continuous keypoints). These continuous annotations are used only for 3D evaluation and do not participate in training.
>
>
>   For the 3D evaluation on these 5058 frames: (1) **For DeepLabCut**, we first obtain 3D keypoints via stereo triangulation from the trained 2D models, then reproject the 3D poses back to the image plane and compute 2D errors against the “ground truth -- continuous keypoints” on all 5058 frames. (2) **For OpenRatEngine**, we fit the 3D virtual rat pose to the multi-view silhouettes on the **same 5058 frames**, and then project the fitted 3D skeleton through the calibrated cameras to obtain 2D keypoints, which are compared to the same “ground truth -- continuous keypoints”.
>
> ### Q2:  Quantitative performance of OpenRatEngine in Table 3
>
> Overall, OpenRatEngine achieves better performance than DeepLabCut (DLC) in Table 3, although for a subset of joints DLC exhibits smaller 2D reprojection errors. This behavior can be explained by two main factors:
>
>
> (1)DLC is trained directly in the 2D image plane for each camera view, using per-view 2D keypoint annotations as supervision, and 3D keypoints are subsequently obtained via triangulation. In Table 3, we report the 2D reprojection error on camera 1, which exactly matches DLC’s native training view and loss definition. In contrast, OpenRatEngine’s 3D pose is obtained by fitting a virtual skeleton to three-view silhouette consistency with IK constraints. This fitting procedure explicitly trades off reprojection accuracy across all three cameras and may sacrifice a few pixels on camera 1 in order to achieve a better global match on cameras 2 and 3. Consequently, evaluating error on a single view is inherently more favorable to DLC and constitutes a stricter test for OpenRatEngine.
>
>
> (2) Stability of the 3D skeleton under occlusion. The primary strength of OpenRatEngine lies in its ability to maintain a stable full-body 3D skeleton, especially for occluded body parts. As shown in Table 3, the limbs, spine, and mid-tail segments are governed by explicit 3D skeletal constraints in OpenRatEngine. For highly occluded behaviors such as locomotion, sniffing, and rearing, this leads to substantially lower reconstruction errors on these joints compared to DLC, which is more prone to large errors or keypoint failures when self-occlusion is severe.
>
> ### Q3: Automatic annotations in the dataset
> The automatically annotated data comprise both **real** and **synthetic** components.
>
>
> (1) For the **real data**, among the 609K frames we manually annotated 6558 frames, and used these to train a DeepLabCut model. The trained model was then applied to the remaining 602K frames to automatically predict 2D keypoints, which we treat as **pseudo-labels** and can directly use for temporal prediction tasks.
>
>
> (2) For the **synthetic data**, each motion sequence generated by OpenRatEngine is driven by the same underlying 3D skeleton and joint set. As a result, any synthetic motion sequence can be exported from Blender in a **uniform format as a complete 3D keypoint trajectory**, effectively yielding synthetic data that are “born with” precise 3D annotations.

---

> > ### Author Response · Authors · 2025-11-21
> > **Author Response to Reviewer gBuZ**
> >
> > ### Q4: Occlusions and camera configuration
> >
> >
> > Compared to Rat7M, our dataset uses fewer camera views, which indeed leads to more self-occlusions in the raw videos. However, the camera configuration in ActionRat is deliberately chosen to better reflect realistic experimental and application settings, and is therefore more representative in practice. In contrast, Rat7M offers a multi-view setup with relatively few occlusions, corresponding more closely to an “ideal” condition. Our goal is to advance fine-grained behavior analysis methods under more constrained camera configurations (down to a single view), rather than relying on heavily instrumented setups. To this end, we developed the virtual rat engine **OpenRatEngine**, which can produce consistent, fully detailed 3D skeleton sequences from arbitrary viewpoints—even in the presence of occlusions.
> >
> >
> > ### Q5: Triangulation
> >
> > We did experiment with using all three cameras jointly for multi-view triangulation. However, under the current occlusion patterns and 2D detection noise levels in ActionRat, three-view triangulation yielded only marginal improvements in MPJPE/MPJVE, while requiring additional outlier-rejection mechanisms for the third view. In practice, this sometimes introduced jitter and instability for certain joints and substantially increased implementation complexity. In this work, we therefore let DeepLabCut and OpenRatEngine each operate with their most suitable 3D pipeline, and then project the resulting 3D keypoints back onto the same camera view, where we compare them against the corresponding 2D human annotations and compute reprojection error. The dataset already includes full three-camera calibration files, and we encourage future work to explore more advanced multi-view triangulation or optimization algorithms on ActionRat.
> >
> >
> > ### Q6: Why are the two ears not modeled in the OpenRatEngine model?
> > Our current work is primarily concerned with sensorimotor control, with a particular focus on the limbs and spine. From the perspective of rat physiology, there are no articulating joints at the ears; accordingly, in OpenRatEngine we treat the ears as part of the rigid head segment rather than defining separate movable bones. For this reason, in Table 3 we only compare DeepLabCut and OpenRatEngine on the **10 shared keypoints excluding the ears**, in order to ensure a fair alignment and comparison between the two methods.

---

> > > ### Comment · Reviewer_gBuZ · 2025-11-25
> > >
> > > Thank you for taking the time to respond to my concerns.
> > >
> > > Some variant of the responses to Q1 and Q3 should be added to the paper, to clarify the role of the 1500 vs 5058 frames, as this is crucial for interpreting both the results of the paper and what data is available.
> > >
> > > Are the 6558 frames in Q3 the same frames as the 1500 + 5058 frames in Q1?
> > >
> > > I do believe that OpenRatEngine should in principle provide more stable estimations over time and with occlusions, and has likely been a lot of work in putting together.
> > > However, I still feel that the paper does not really showcase this model well. Table 4 does not really represent the advantage of the OpenRatEngine, as OpenRatEngine does not consistently do better than DeepLabCut so it's hard to compare at a glance. I wonder if the authors could add at least an average error for each model within each action?
> > >
> > > The rest of my questions are answered. I keep my rating.

---

> > > > ### Author Response · Authors · 2025-12-03
> > > > **Thanks for the feedback**
> > > >
> > > > We are grateful for your constructive suggestions, which have led to a clearer and more informative presentation of both the ActionRat dataset and the role of OpenRatEngine in our work. We also greatly appreciate your latest feedback on the scores.

---

### Author Response · Authors · 2025-11-21
**Common Concern from reviewers**

### Common Concern (from reviewers Wxag and ZMKX): data scale and data value


We would like to clarify that the core distinction between our dataset and existing public resources lies in its application value rather than sheer size. As also emphasized by reviewer JWJZ, the value of a dataset should not be judged solely by technical novelty or the number of frames, but by the uniqueness of the tasks it enables and the granularity of its annotations. We summarize our perspective as follows.


(1) **Necessity of ActionRat dataset**. As shown in Table 1, conventional video datasets only require recording RGB videos and extracting 2D/3D poses, and therefore cannot probe the “**Perception - Movement**” mechanisms by which the brain drives movement through stimulation. In contrast, ActionRat tightly aligns **video–neural implant sites–stimulation parameters**: each behavioral segment is associated with metadata such as stimulation channel, pulse frequency, and voltage amplitude. To the best of our knowledge, ActionRat is the only dataset that provides a complete “video + electrical stimulation parameters + fine-grained actions” pipeline, making it uniquely suited for studying nerve stimulation–behavior relationships.


(2) **Behavioral diversity and fine-grained annotation**. The 7 fine-grained sub-actions annotated in ActionRat differ from standard coarse labels and are defined based on **rat behavioral semantics**. Any movement sequence can be decomposed into a composition of these sub-actions; for example, a trajectory from A to B may be annotated as “locomotion → turning → sniffing → pausing”. This design allows ActionRat to support analyses of **trajectory variability under the same semantic action** as a function of stimulation. In the context of brain-computer interfaces (BCI ), ActionRat thus supports not only standard 3D pose estimation and temporal modeling, but also, for the first time, tasks such as stimulation–behavior mapping and quantitative evaluation of BCI-controlled movement quality.

---

### Author Response · Authors · 2025-12-03
**Response to the AC and Reviewers**

Dear AC and Reviewers,

Thank you again for your time and multiple rounds of communication, which have significantly helped improve the quality of our work. We have carefully addressed your main concerns in detail and updated the pdf manuscript accordingly. We hope that our responses adequately address your comments.

Best, Authors

---

### Meta-Review · Area_Chair_18nf · 2026-01-04

**Summary:**

Reviewers didn’t think the topic was uninteresting. The problem is that the paper doesn’t clearly earn its place as an ICLR datasets-and-benchmarks contribution. Several reviewers felt the “why this dataset matters” story is still weak compared with existing resources (especially Rat7M): it’s not obvious what new failure modes it exposes, what genuinely new tasks it enables, or what insights are hard to get elsewhere. OpenRatEngine was also seen as a reasonable integration of standard components and existing tools, but the experiments don’t convincingly show the claimed benefits (occlusion robustness, synthetic data helping, or real value from the BCI metadata). On top of that, parts of the experimental setup and tables were confusing, and reviewers wanted clearer definitions and variance/significance to interpret the results. With two reviewers firmly unconvinced on these core points, the discussion leaned toward reject despite a couple more positive takes.

**Reviewer Concerns:**

The rebuttal did a solid job clearing up confusion and misunderstandings. It clarified how the annotated frames are used, how the 3D evaluation is done, where the keypoints come from, and gave more concrete explanations for modeling choices like silhouette fitting, triangulation, and the design of OpenRatEngine. However, the core concerns didn’t really go away. Reviewers are still not convinced that the dataset clearly enables new or better ML results, especially when it comes to actually using the BCI metadata. The diversity and “more challenging benchmark” claims remain unpersuasive without stronger analysis and clearer comparisons, and the experiments still don’t clearly demonstrate the main advantages the paper argues for, such as synthetic data meaningfully helping or improved robustness and generalization. Overall, the rebuttal improved clarity, but it didn’t close the evidence gap that the negative reviewers are focused on.

**Reviewer Scores:**

gBuZ (8) would almost certainly stay at 8: they asked for clarification and presentation changes, accepted the answers, and explicitly said they keep their rating. JWJZ (6) would likely stay at 6: they view it as interesting and worthwhile despite limited technical novelty, and they also stated they maintain the rating. Wxag (2) would remain at 2: after the rebuttal they reiterated that the dataset value is still not demonstrated, questioned significance and variance, and restated the “Blender dependence” concern. ZMKX (2) would also remain at 2: their critique is fundamentally about novelty, task relevance, and lack of convincing downstream demonstrations and generalization tests, and the rebuttal does not appear to have shifted that core stance. pF94 (2) is very likely to remain at 2 as well: even after extensive answers, they continued to challenge the diversity claims, the cross-dataset difficulty argument, and the evaluation choices, and they are asking for additional analyses that are not resolved within the discussion.

---

### Decision · Program_Chairs · 2026-01-26

Reject